# Video Scene Segmentation with Genre and Duration Signals

**Jungu Cho**[1,2]**, Seong Jong Ha**[1]**, Hae-Gon Jeon**[3*]
[1]AI R&D Division, CJ Corporation
[2]AI Graduate School, GIST
[3]Department of Artificial Intelligence, Yonsei University

## Abstract

Video scene segmentation aims to detect semantically coherent boundaries in long-form videos, bridging the gap between low-level visual signals and high-level narrative understanding. However, existing methods primarily rely on visual similarity between adjacent shots, which makes it difficult to accurately identify scene boundaries, especially when semantic transitions do not align with visual changes. In this paper, we propose a novel approach that incorporates production-level metadata, specifically genre conventions and shot duration patterns, into video scene segmentation. Our main contributions are three-fold: (1) we leverage textual genre definitions as semantic priors to guide shot-level representation learning during self-supervised pretraining, enabling better capture of narrative coherence; (2) we introduce a duration-aware anchor selection strategy that prioritizes shorter shots based on empirical duration statistics, improving pseudo-boundary generation quality; (3) we propose a test-time shot splitting strategy that subdivides long shots into segments for improved temporal modeling. Experimental results demonstrate state-of-the-art performance on MovieNet-SSeg and BBC datasets. We introduce MovieChat-SSeg, extending MovieChat-1K with manually annotated scene boundaries across 1,000 videos spanning movies, TV series, and documentaries.

## 1 Introduction

Understanding long-form video content requires structuring it into semantically meaningful units that reflect the underlying narrative. Unlike short clips, narrative-driven videos such as movies, TV shows, and documentaries are naturally organized into distinct scenes that maintain narrative coherence. Segmenting such content into scenes provides a useful abstraction that supports a wide range of downstream tasks, including content-based retrieval (Bain et al. (2020; 2021)), summarization (Gan et al. (2023); Argaw et al. (2024)), and multi-modal question answering (QA) (Song et al. (2024); Chandrasegaran et al. (2024)). However, current approaches (Chen et al. (2021); Wu et al. (2022); Mun et al. (2022); Islam et al. (2023); Yang et al. (2023); Chen et al. (2024)) predominantly use visual features to identify scene boundaries, leading to errors when narrative transitions occur without accompanying visual discontinuities. These limitations highlight the need to move beyond purely visual approaches toward methods that can leverage richer contextual information.

Professional video production offers rich contextual cues through deliberately crafted elements such as editing patterns, camera movements, and temporal structures that signal narrative transitions. Among these production elements, genre conventions and shot duration patterns offer particularly promising signals for computational scene analysis. However, leveraging these signals presents several challenges: genre information is typically available only at the video level, while shot duration patterns vary significantly across genres and production styles. Addressing these challenges requires new methods that can integrate production metadata into scene segmentation while handling their inherent variability and granularity mismatches.

We propose to encode genre conventions through textual descriptions and use them as semantic priors for shot representation learning. Our framework integrates this genre information during

---
*Corresponding author

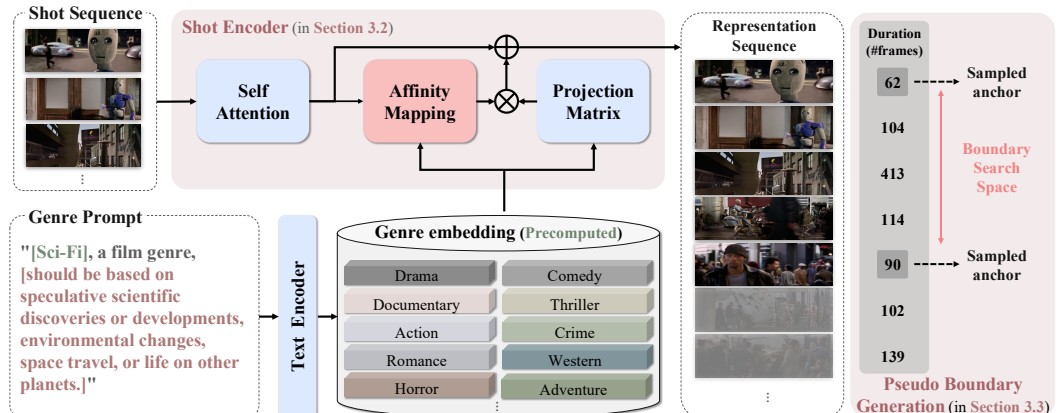

Figure 1: Our framework integrates genre embeddings into the shot encoder via affinity-based residual connections and uses duration-aware anchor sampling to generate pseudo-boundaries for self-supervised shot representation learning in scene segmentation.

pretraining to enhance shot-level representations. We construct textual prompts from IMDb-based genre definitions and compute affinities with visual features to provide semantic guidance. This approach aims to capture semantic relationships that may not be apparent from visual features alone.

Building on prior self-supervised learning frameworks based on pseudo scene boundaries, we introduce a duration-aware anchor sampling strategy that leverages shot duration patterns to improve representation learning. Our approach prioritizes shorter shots as anchors to provide more diverse training samples, as these shots are more frequently distributed across different temporal contexts within sequences. Specifically, we model the empirical shot duration distribution within each video and assign higher sampling weights to shots in the shorter duration percentiles during anchor pair selection for pseudo-boundary generation. This strategy enables the formation of diverse subsequences within fixed-length sequences, providing varied training samples compared to fixed anchor selection methods (Mun et al. (2022); Islam et al. (2023)).

We propose a test-time shot split strategy to handle long shots containing multiple semantic segments during inference. Our strategy subdivides shots exceeding a duration threshold (10 seconds) into smaller segments, each represented by a keyframe. This preprocessing approach requires no model retraining and can be applied to existing scene segmentation frameworks.

We introduce *MovieChat-SSeg*, a scene segmentation benchmark derived from MovieChat-1K Song et al. (2024) with 1,000 annotated videos. The dataset contains manually annotated scene boundaries, created through a multi-annotator process. The dataset includes diverse content types spanning movies, TV series, and documentaries. Each video represents a segment extracted from longer content, enabling evaluation across different narrative contexts.

Our main contributions are summarized as follows:

- We incorporate genre conventions as semantic priors by encoding textual definitions from IMDb guidelines, enabling genre-guided shot representation learning during pretraining.

- We introduce a duration-aware anchor sampling strategy that prioritizes shorter shots based on empirical duration distributions, generating more diverse pseudo-boundaries compared to fixed-anchor approaches.

- We propose a test-time shot splitting strategy that subdivides long shots into segments during inference, requiring no model retraining or architectural changes.

- We introduce MovieChat-SSeg, a scene segmentation benchmark with 1,000 manually annotated videos spanning films, TV series, and documentaries.

## 2 RELATED WORKS

### 2.1 VIDEO SCENE SEGMENTATION

Video scene segmentation typically begins with shot boundary detection (Castellano (2014); Soucek & Lokoc (2024)) to identify basic temporal units. Traditional methods group consecutive shots into scenes using low-level visual features such as color histograms (Rui et al. (1999); Chasanis et al. (2008)) and motion patterns (Rasheed & Shah (2005)). However, these feature-based approaches often fail to capture higher-level semantic relationships between shots, limiting their effectiveness on complex narrative content.

The release of MovieNet (Huang et al. (2020)) provides a substantial benchmark for scene segmentation research with annotations across multiple movie genres. Supervised approaches (Huang et al. (2020); Rao et al. (2020a)) using this dataset achieve strong performance but require extensive manual annotations, limiting their applicability to large-scale unlabeled video collections. To overcome this limitation, recent works adopt self-supervised learning frameworks that leverage unlabeled videos. These approaches (Mun et al. (2022); Islam et al. (2023); Yang et al. (2023)) typically follow a two-stage paradigm: (i) pretraining a shot encoder on an unlabeled large-scale dataset using pseudo-boundaries generated by pretext tasks, and (ii) fine-tuning on a smaller labeled dataset.

Various pretext tasks have been proposed for self-supervised scene segmentation, including methods based on shot similarity (Chen et al. (2021)), scene consistency (Wu et al. (2022)), and metadata-based movie similarity (Chen et al. (2023)). BaSSL Mun et al. (2022) introduces a method to generate pseudo scene boundaries by using the two fixed anchor shots located at the ends of a given shot sequence. The model then learns contextual representations by encoding the relationships across shots within the sequence. Subsequent approaches extend this framework by incorporating long-range inter-shot relationships (Islam et al. (2023)) and multi-scale temporal context (Yang et al. (2023)). While these methods show promising results, most rely primarily on visual and temporal cues, with limited exploration of comprehensive production-level metadata for semantic guidance.

Movies2Scenes (Chen et al. (2023)) explores the use of metadata tags, such as co-watch patterns, genre labels, and synopsis text, for scene grouping. Specifically, it measures movie-level similarity using these metadata and constructs positive scene pairs across movies with similar tags, which are used to train a contrastive learning objective. This work demonstrates the potential of leveraging metadata for scene-level analysis, focusing on inter-movie metadata comparison.

VSS-MGP (Bouyahi & Ayed (2020)) employs multimodal signals to train a classification model for the four most frequent genres, then performs scene segmentation by computing similarity between adjacent shots based on their shot-level genre prediction distributions. Movie-CLIP (Zhang et al. (2024)) proposes a multi-modal fusion architecture originally designed for genre prediction and shows this structure can be applied to scene segmentation. However, VSS-MGP (Bouyahi & Ayed (2020)) requires shot-level genre predictions, while Movie-CLIP (Zhang et al. (2024)) does not explicitly utilize genre information for segmentation.

Beyond visual features, shot duration patterns have been recognized in film studies as indicators of narrative structure (Reisz & Millar (1971); Salt (2009)). However, the considerable variation in duration distributions across genres and production styles (Cutting & Candan (2015)) makes it challenging to develop consistent supervision strategies, motivating the need for adaptive approaches.

### 2.2 LONG-FORM VIDEO UNDERSTANDING

Understanding long-form videos, including video question answering (Tapaswi et al. (2016); Song et al. (2024)) and narrative summarization (Song et al. (2015); Rohrbach et al. (2015)), requires modeling semantic structures over extended temporal ranges. These tasks typically operate on higher-level narrative units, such as scenes or events, rather than frame-level analysis. Scene segmentation can provide useful structural units for such tasks by defining semantically coherent segments for temporal reasoning and content organization.

Recent datasets for long-form video understanding, such as MovieChat-1K (Song et al. (2024)) and TVQA (Lei et al. (2018)), incorporate textual annotations including subtitles and dialogue to support multimodal reasoning tasks. However, many of these datasets lack explicit scene boundary annota-

tions, limiting opportunities to study scene-level video organization. For instance, MovieChat-1K focuses on multimodal QA tasks and does not include scene boundary annotations.

## 3 METHOD

### 3.1 TASK DEFINITION

Narrative-driven videos, such as movies and TV shows, are typically structured into a four-level hierarchy: **frames**, **shots**, **scenes**, and the whole **video**. A *shot* refers to a sequence of consecutive frames captured without interruption by a single camera, and can be reliably identified using low-level visual cues (Cotsaces et al. (2006); Castellano (2014)). In contrast, a *scene* is a semantically cohesive unit composed of multiple shots that maintain continuity in action, location, and time (Katz (1979)), serving as a key structural element of narrative storytelling.

While shot boundaries are often associated with visual discontinuities, detecting scene boundaries requires understanding higher-level semantic relationships, such as thematic coherence, location changes, and temporal continuity. Video scene segmentation aims to identify these boundaries by determining whether a given shot marks the end of a scene. This can be formulated as a binary classification problem for each shot, where the model must integrate visual features with contextual information from surrounding shots to make boundary predictions.

Formally, a video can be represented as a sequence of $N$ shots $\{s_1, s_2, \ldots, s_N\}$ with corresponding binary boundary labels $\{y_1, y_2, \ldots, y_N\}$. Here, $y_i \in \{0, 1\}$ indicates whether $s_i$ marks the end of a scene. Each shot $s_i$ is transformed into a feature vector $e_i$ by a shot encoder $\theta_E$. These shot representations serve as the input for boundary prediction, enhanced by auxiliary semantic signals described in the following sections.

### 3.2 GENRE-GUIDED SHOT REPRESENTATION

Genre conventions influence the visual and narrative elements of scenes, potentially providing semantic context for understanding scene structures. While visual appearances may vary across shots, genre conventions may provide consistent semantic signals that could aid in identifying narrative coherence within scenes. However, applying genre directly is nontrivial, as most movies are associated with multiple genre tags, making it difficult to assign a unique genre to each shot or scene.

To address this, we utilize genre conventions as a soft semantic prior during the self-supervised pre-training stage. We construct genre-specific textual prompts based on IMDb definitions (IMDb.com (2023)), incorporating characteristic visual and narrative descriptions for each genre. These prompts are encoded into embedding vectors using the text encoder from CLIP (Radford et al. (2021)), forming a genre embedding set $G_e \in \mathbb{R}^{N_g \times D}$, where $N_g$ is the number of genre categories and D is the embedding dimension.

Our shot encoder incorporates pre-computed genre embeddings as fixed parameters within the ViT (Dosovitskiy et al. (2020)) architecture. Within the ViT layers, we compute affinities between visual token features $V_t \in \mathbb{R}^{N_t \times D}$ and the stored genre embeddings $G_e$ using cosine similarity. The resulting affinity matrix A represents the relevance of each genre to the visual tokens. The genre embeddings are then projected and integrated with the original visual features via residual connections:

$$A = \text{softmax}\left(\frac{V_t G_e^T}{\sqrt{D}}\right), \quad V_t^{\text{genre}} = V_t + A G_e W, \tag{1}$$

where $W \in \mathbb{R}^{D \times D}$ projects the weighted genre information before integration. Following the ViT architecture, the enhanced token features $V_t^{genre}$ are aggregated to form the final shot-level representation $e_i$.

### 3.3 DURATION-AWARE PSEUDO-BOUNDARY GENERATION

Existing pseudo-boundary generation methods, particularly BaSSL (Mun et al. (2022)), use fixed anchor positions at sequence endpoints, limiting their ability to capture diverse temporal patterns in video sequences. We observe that scenes often exhibit consistent shot duration patterns, with

many scenes dominated by shorter shots that form coherent narrative units. To leverage this insight, we modify the anchor selection mechanism by introducing a duration-aware sampling strategy that prioritizes shorter shots over fixed endpoint selection.

Instead of selecting fixed anchors, we impose higher sampling probabilities based on inverse-normalized shot durations within a sequence. We sample anchor shots $\{s_l, s_r\}$ from the subsequences on both sides of the target shot based on the sampling probability $P(s_i)$:

$$P(s_i) = \frac{1/d_i}{\sum_{j=1}^{N} 1/d_j},\tag{2}$$

where $d_i$ represents the duration of the $i$-th shot, and $N$ is the total number of shots in the subsequence. We then search for a pseudo-boundary $b^*$ between these anchor shots using a similarity-based approach adapted from Dynamic Time Warping (DTW) (Berndt & Clifford (1994)):

$$b^*(i,l,r) = \operatorname*{argmax}_{b=-L,...,R-1} \left( \frac{1}{b+L+1} \sum_{j=-L}^{b} \mathrm{sim}(e_l, e_{i+j}) + \frac{1}{-b+R} \sum_{j=b+1}^{R} \mathrm{sim}(e_r, e_{i+j}) \right), \tag{3}$$

where $\mathrm{sim}(x,y)$ is cosine similarity between shot representations, $i$ is the target shot index, and $L$, $R$ denote distances to left and right anchors respectively. This process divides the input sequence into two pseudo-scene sequences: $S_i^l = \{s_{i-L}, ..., s_{b^*}\}$ and $S_i^r = \{s_{b^*+1}, ..., s_{i+R}\}$.

### 3.4 TEST-TIME SHOT SPLIT STRATEGY

While our training methods enhance shot representation learning, real-world inference faces challenges from highly variable shot durations. Long shots may contain multiple semantic segments that are difficult to process as single units, potentially affecting boundary detection accuracy in sequence-based approaches. To address this challenge, we propose a test-time shot split strategy that subdivides long shots during inference without requiring model retraining.

For any shot $s_i$ with duration $d_i > \tau$ (set to 10 seconds), we partition it into three equal-length segments, each treated as an independent shot. When pre-extracted keyframes are available (e.g., three keyframes per shot in MovieNet (Huang et al. (2020))), we directly assign them to the segments; otherwise, we extract keyframes from each segment using uniform temporal sampling:

$$\{s_i^{(1)}, s_i^{(2)}, s_i^{(3)}\} = \mathrm{split}(s_i) \quad \text{if } d_i > \tau \tag{4}$$

The strategy operates at the preprocessing level, making it compatible with existing architectures. After boundary prediction, detected boundaries are mapped back to original temporal coordinates.

## 4 MOVIECHAT-SSEG DATASET

We present **MovieChat-SSeg**, a manually annotated benchmark for the evaluation of video scene segmentation. The dataset builds upon the publicly available *MovieChat-1K* (Song et al. (2024)) dataset, which contains 1,000 video clips originally curated for long-form video-language tasks. We extend these clips by manually annotating scene boundaries to create a testbed for scene segmentation research.

**Annotation Process.** Scene boundaries are annotated by trained annotators following established guidelines that consider narrative coherence, location changes, and temporal transitions. The annotation process involves multiple stages: initial independent annotation, discussion of boundary cases, and consensus resolution for disagreements. All annotations are reviewed to ensure consistency across the dataset.

**Dataset Characteristics.** MovieChat-SSeg complements existing benchmarks by including diverse content types spanning movies, TV series, and documentaries. This content variety enables evaluation across different narrative structures and editing styles. This diversity enables evaluation of scene segmentation methods across varied editing styles and storytelling approaches.

The dataset consists of focused segments (average 7.4 minutes) that enable efficient annotation while capturing multiple scene transitions per video. However, this focus on shorter segments may not capture long-term narrative patterns present in full-length content.

We release MovieChat-SSeg publicly with annotation guidelines and evaluation protocols to support future research.

| Dataset | #Title | #Video | Time(h) | #Shot | #Scene | Category |
|---------|--------|--------|---------|-------|--------|----------|
| BBC | 11 | 11 | 9 | 4.9K | 547 | Documentary |
| MovieNet-SSeg(Test) | 64 | 64 | 132 | 106K | 8.1K | Movie |
| MovieChat-SSeg(Ours) | 143 | 1K | 124 | 116K | 7.8K | Movie, Documentary, TV series |

Table 1: Comparison of video scene segmentation dataset characteristics.

## 5 EVALUATIONS

### 5.1 DATASET AND EVALUATION METRICS

We use MovieNet (Huang et al. (2020)) with 1,100 movies for pre-training. For supervised training and evaluation, we utilize the MovieNet-SSeg subset, which includes 318 movies with scene boundary annotations, following the official training(190), validation(64), and test(64) splits. To assess the generalization ability of our model, we conduct zero-shot evaluation on both the BBC (Baraldi et al. (2015)) dataset and our newly proposed MovieChat-SSeg dataset.

We evaluate model performances using Average Precision (AP), AUC-ROC, and F1 score for shot-level classification, and mean Intersection over Union (mIoU) for sequence-level segment alignment.

### 5.2 TRAINING STRATEGY

Following prior works (Mun et al. (2022); Islam et al. (2023)), we adopt a modular architecture consisting of a shot encoder, a context encoder $\theta_c$, and a prediction head $h_p$. The context encoder models temporal dependencies across shots, and the prediction head outputs binary scene boundary predictions. During pretraining, the shot encoder is optimized using pseudo-boundaries and then frozen during fine-tuning, where the context encoder and prediction head are updated only.

We employ two objective functions in the pretraining stage. First, we apply a contrastive loss based on InfoNCE (Oord et al. (2018)), where the average representation of shots within a pseudo scene is treated as a scene-level feature, and paired with an anchor shot from the same scene, as follows:

$$L_{\text{nce}}(e, \bar{e}) = -\log \frac{exp(\text{sim}(e, \bar{e}))}{exp(\text{sim}(e, \bar{e})) + \sum_{e_n} exp(\text{sim}(e_n, \bar{e})) + \sum_{\bar{e}_n} exp(\text{sim}(e, \bar{e}_n))}, \quad (5)$$

where $e$ denotes the anchor shot representation, and $\bar{e}$ represents the average feature of a pseudo scene. $e_n$ and $\bar{e}_n$ represent negative shot representations and negative pseudo-scene features, respectively. Since each shot sequence is divided into two pseudo scenes by a pseudo boundary at index $b^*$, we compute the contrastive loss for both segments: $L_{con} = L_{nce}(e_l, \bar{e}_l) + L_{nce}(e_r, \bar{e}_r)$, where $e_l$ and $e_r$ are the anchor representations for the left and right pseudo scenes, respectively. $\bar{e}_l = \text{avg}(e_l, ..., e_{b^*})$ and $\bar{e}_r = \text{avg}(e_{b^*+1}, ..., e_r)$ denote the average scene-level features of each segment. Second, we apply a binary cross-entropy loss to distinguish between the pseudo-boundary shot and a non-boundary shot randomly selected from the given sequence as below:

$$L_{\text{pb}}(c_{b^*}, c_n) = -\log(h_p(c_{b^*})) - \log(1 - h_p(c_n)), \quad (6)$$

where $c_{b^*}$ and $c_n$ denote context-aware shot representations of the pseudo-boundary and a non-boundary shot, respectively, obtained from the context encoder. The function $h_p(\cdot)$ denotes the prediction head, which maps the input representation to the probability of a boundary. The overall objective in the pretraining stage is defined as the linear combination of the contrastive loss and the binary cross-entropy loss for pseudo-boundary prediction.

During fine-tuning, we train the context encoder and prediction head using scene boundary labels with a standard binary cross-entropy loss as follows:

$$L_{\text{sb}}(c_i, y_i) = -y_i \log(h_p(c_i)) + (1 - y_i) \log(1 - h_p(c_i)), \quad (7)$$

where $c_i$ is the context-aware representation of the $i$-th shot and $y_i \in \{0, 1\}$ is the ground truth scene boundary label. At inference time, we predict a shot as a scene boundary if the boundary probability $h_p(c_i)$ is greater than or equal to 0.5.

| Method | AP | AUC | F1 | mIoU |
|---|---|---|---|---|
| GraphCut (Rasheed & Shah (2005)) | 14.10 | - | - | 20.70 |
| SCSA (Chasanis et al. (2008)) | 14.70 | - | - | 30.50 |
| DP (Han & Wu (2011)) | 15.50 | - | - | 32.00 |
| SceneTiling* (Wang et al. (2025)) | 19.95 | 72.15 | 29.36 | 36.96 |
| Story Graph (Tapaswi et al. (2014)) | 25.10 | - | - | 35.70 |
| Grouping (Rotman et al. (2016)) | 33.60 | - | - | 37.20 |
| Siamese (Baraldi et al. (2015)) | 35.80 | - | - | 39.60 |
| MS-LSTM (Huang et al. (2020)) | 46.50 | - | - | 46.20 |
| LGSS (Rao et al. (2020b)) | 47.10 | - | - | 48.80 |
| Movie-CLIP (Zhang et al. (2024)) | 54.45 | - | - | - |
| MHRT (Wei et al. (2023)) | 54.80 | 90.30 | 46.30 | 51.20 |
| ShotCoL (Chen et al. (2021)) | 53.40 | - | - | - |
| SCRL (Wu et al. (2022)) | 54.82 | - | 51.43 | - |
| Movies2Scenes (Chen et al. (2023)) | 55.03 | - | - | - |
| CMS (Park et al. (2024)) | 55.73 | - | 52.05 | - |
| BaSSL (Mun et al. (2022)) | 57.40 | 90.54 | 47.02 | 50.69 |
| CAT (Yang et al. (2023)) | 59.55 | 91.81 | 51.93 | 53.67 |
| TranS4mer (Islam et al. (2023)) | 60.78 | 91.89 | 48.36 | 51.91 |
| Ours | **63.62** | **92.84** | **58.88** | **59.64** |

(a) Results on the MovieNet-SSeg.

| Method | A1 | A2 | A3 | A4 | A5 | Avg |
|---|---|---|---|---|---|---|
| ShotCoL | 29.9 | 30.8 | 31.5 | 26.5 | 21.3 | 28.0 |
| SCRL | 32.5 | 32.5 | 33.3 | 28.4 | 24.3 | 30.2 |
| BaSSL* | 33.7 | 30.6 | 32.3 | 26.1 | 22.2 | 29.0 |
| TranS4mer* | 35.6 | 33.6 | 29.8 | 27.6 | 25.0 | 30.3 |
| Ours | **42.2** | **42.1** | **42.8** | **32.8** | **26.3** | **37.2** |

(b) Results on the BBC. The table presents Average Precision (AP) scores computed against ground truth annotations from five independent expert annotators.

| Method | Movie | TV | Docu. | Total |
|---|---|---|---|---|
| BaSSL* | 37.6 | 43.1 | 23.3 | 36.6 |
| TranS4mer* | 38.8 | 46.7 | 24.4 | 37.9 |
| Ours | **48.1** | **54.6** | **29.4** | **46.7** |

(c) Results on the MovieChat-SSeg.

Table 2: Comparison results across three video scene segmentation benchmark datasets. In the (b) BBC dataset and (c) MovieChat-SSeg dataset, evaluation results are presented using Average Precision (AP). The best numbers are in **bold**. The second best is in underline.(*our implementation)

## 5.3 IMPLEMENTATION DETAILS

We use ViT-B/32 (Dosovitskiy et al. (2020)) as the shot encoder backbone, initialized with Open-CLIP (Cherti et al. (2023)) weights, processing 224×224 resolution images with 32×32 patch size. We freeze the pre-trained ViT backbone weights and train only the genre integration module (projection matrix W) and the final MLP classification head. Following (Huang et al. (2020); Zhang et al. (2024)), we use 21 genre categories with textual definitions from IMDb (IMDb.com (2023)), encoded using the OpenCLIP text encoder. For sequence-level boundary prediction, we employ a two-layer BERT Devlin et al. (2018) architecture with 768 hidden dimensions, trained from scratch.

We adopt a two-stage training strategy. During pretraining, we train the shot and context encoders for 20 epochs with a batch size of 256, using a base learning rate of 0.3 with cosine decay and linear warmup for 1 epoch. Fine-tuning is performed for 20 epochs using a batch size of 1,024. We use the Adam (Kingma (2014)) optimizer with a learning rate of $2.5 \times 10^{-6}$ and a momentum of 0.9. All models are trained on eight A100 GPUs. Results are averaged over five runs with different fine-tuning seeds.

## 5.4 COMPARISON RESULTS

**Results on the MovieNet-SSeg Dataset** We compare our model with the state-of-the-art methods on MovieNet-SSeg, and report the results in Tab. 2(a). Pseudo-boundary-based self-supervised methods (Mun et al. (2022); Yang et al. (2023); Islam et al. (2023)) achieve strong performance compared to traditional approaches. Our method shows consistent improvements over existing approaches using various backbone architectures, including significant gains over methods employing the same ViT backbone (Islam et al. (2023)). This demonstrates that our contributions of genre-guided representation learning and duration-aware sampling provide complementary benefits beyond architectural choices. Qualitative analysis in Fig. 2 illustrates improved boundary detection in challenging scenarios.

**Cross-domain Evaluation on BBC Dataset** For the cross-domain evaluation, we conduct an additional experiment on the BBC dataset (Baraldi et al. (2015)). The BBC dataset contains nature and wildlife documentaries with different narrative structures from MovieNet's theatrical content. The dataset provides multiple expert annotations for the same videos, enabling evaluation against diverse boundary interpretations. Our method achieves competitive performance compared to existing approaches, as shown in Tab. 2(b). These results suggest that our approach generalizes reasonably well to documentary content beyond theatrical films.

| Index | Method | Backbone | Weight Init. | Train. Params | Anchor Select. | Split Thre. | MovieNet-SSeg | | | | BBC(AP) | MovieChat-SSeg(AP) | | | |
|---|---|---|---|---|---|---|---|---|---|---|---|---|---|---|---|
| | | | | | | | AP | AUC | F1 | mIoU | | Movie | TV | Docu. | Total |
| 1 | Ours | ViT-B/32 | CLIP | 39M | Side | – | 62.82 | 92.39 | 58.00 | 59.15 | 36.11 | 46.91 | 53.28 | 28.35 | 43.94 |
| 2 | BaSSL* | ViT-B/32 | CLIP | 102M | Side | – | 60.40 | 91.46 | 45.76 | 50.04 | 31.12 | 38.69 | 45.83 | 29.40 | 38.05 |
| 3 | TranS4mer* | ViT-B/32 | CLIP | 125M | Side | – | 60.32 | 91.44 | 47.04 | 50.54 | 29.40 | 38.83 | 46.57 | 29.88 | 38.34 |
| 4 | BaSSL* | ResNet-50 | scratch | 39M | Side | – | 56.82 | 90.00 | 51.14 | 54.42 | 29.83 | 37.39 | 42.51 | 22.72 | 34.73 |
| 5 | BaSSL* | ResNet-50 | scratch | 39M | IDW | – | 57.65 | 90.47 | 43.82 | 48.87 | 35.18 | 44.69 | 49.72 | 24.31 | 40.27 |
| 6 | TranS4mer* | ViT-S/32 | Imagenet | 43M | Side | – | 58.60 | 90.92 | 47.07 | 50.29 | 29.68 | 33.76 | 41.35 | 19.47 | 31.97 |
| 7 | TranS4mer* | ViT-S/32 | Imagenet | 43M | IDW | – | 61.17 | 91.78 | 49.09 | 52.42 | 34.11 | 46.56 | 53.74 | 25.42 | 42.69 |
| 8 | BaSSL* | ResNet-50 | scratch | 39M | Side | 10s | 57.40 | 90.18 | 52.86 | 56.00 | 33.32 | 42.71 | 45.82 | 24.15 | 40.49 |
| 9 | TranS4mer* | ViT-S/32 | Imagenet | 43M | Side | 10s | 59.42 | 91.21 | 46.91 | 50.52 | 32.14 | 39.53 | 41.72 | 23.40 | 37.27 |

Table 3: Performance comparison on video scene segmentation benchmarks. Methods are grouped by backbone architecture, with our contributions (duration-aware anchor sampling and test-time shot splitting) applied to state-of-the-art baselines. (*our implementation)

**Evaluation on MovieChat-SSeg Dataset** MovieChat-SSeg offers a diverse evaluation benchmark with video content from three distinct formats: movies, TV series, and documentaries. Each format differs in narrative structure, editing style, and scene organization, making it valuable for assessing the generality of models across various storytelling conventions. As shown in Tab. 2(c), our model achieves competitive performance across all formats, with an overall improvement of +8.8 points over TranS4mer (Islam et al. (2023)). The improvements are consistent across different content types, suggesting that our combined approach of genre-guided learning and duration-aware sampling provides benefits across varied narrative contexts.

**Comparison with Baseline Methods** To ensure fair comparison, we re-implement BaSSL and TranS4mer with the same ViT-B/32 backbone and CLIP initialization. As shown in Tab. 3, both achieve competitive performance, confirming our gains stem from methodological rather than architectural advantages. Applying our duration-aware sampling to these baselines yields consistent improvements (+0.8-2.1 AP on MovieNet), while test-time splitting provides additional gains (+0.6-0.8 AP on MovieNet) without retraining. These results suggest our contributions show promise for broader applicability.

## 5.5 ABLATION STUDIES

We conduct ablation studies to evaluate our design choices and examine the impact of various configurations. Our analysis focuses on four key components: (1) integration methods for incorporating genre embeddings into shot representation learning (Tab. 4, rows 1-3, 8); (2) prompt design strategies for genre embedding generation (rows 4-5, 8); (3) parameter settings for duration-aware anchor sampling (rows 6-7, 8); (4) threshold configurations for test-time shot splitting across different duration settings (rows 8-11).

**Genre Embedding Integration** To assess the impact of genre embedding integration, we compare four strategies: no genre embedding, frame-level concatenation, token-level concatenation, and our

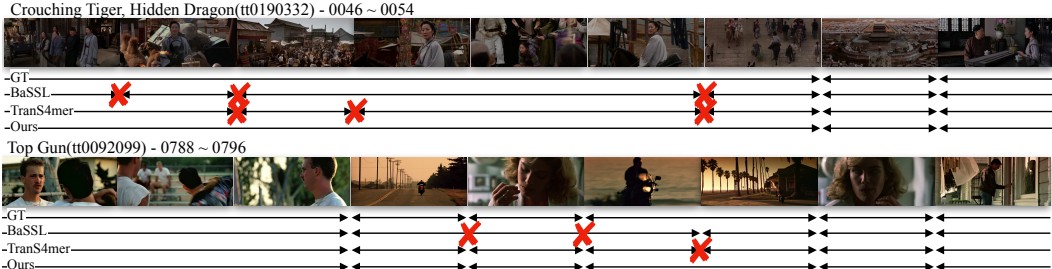

Figure 2: Qualitative examples of scene boundary detection. Top: A visually dynamic single scene where our method avoids over-prediction. Bottom: A cross-cutting sequence where our approach accurately detects boundaries between scene translations. (GT: Ground Truth, BaSSL Mun et al. (2022) and TranS4mer Islam et al. (2023): baseline methods)

| Index | Embedding | Prompt | Anchor | Split Thre. | MovieNet-SSeg | | | | BBC(AP) | MovieChat-SSeg(AP) | | | |
|---|---|---|---|---|---|---|---|---|---|---|---|---|---|
| | | | | | AP | AUC | F1 | mIoU | | Movie | TV | Docu. | Total |
| 1 | w/o Embed | Def | IDW | – | 61.27 | 91.99 | 57.55 | 58.61 | 32.16 | 46.33 | 51.24 | 26.15 | 42.03 |
| 2 | Frame Concat | Def | IDW | – | 59.13 | 91.36 | 55.29 | 57.08 | 32.25 | 44.25 | 48.27 | 23.22 | 39.47 |
| 3 | Token Concat | Def | IDW | – | 62.43 | 92.41 | 58.34 | 59.04 | 36.90 | 47.73 | 53.69 | 28.07 | 44.26 |
| 4 | AG-Residual | Name | IDW | – | 63.23 | 92.68 | 59.05 | 59.48 | 36.59 | 46.96 | 53.36 | 28.75 | 44.08 |
| 5 | AG-Residual | Style | IDW | – | 63.42 | 92.75 | 58.76 | 59.56 | 36.39 | 47.09 | 53.10 | 26.75 | 43.23 |
| 6 | AG-Residual | Def | Side | – | 62.82 | 92.39 | 58.00 | 59.15 | 36.11 | 46.91 | 53.28 | 28.35 | 43.94 |
| 7 | AG-Residual | Def | DW | – | 63.44 | 92.73 | 59.07 | 59.85 | 36.69 | 47.61 | 54.30 | 28.84 | 44.39 |
| 8 | AG-Residual | Def | IDW | – | 63.62 | 92.84 | 58.88 | 59.64 | 37.22 | 48.07 | 54.57 | 29.41 | 46.69 |
| 9 | AG-Residual | Def | IDW | 60s | 63.63 | 92.83 | 58.97 | 59.86 | 37.28 | 48.15 | 54.58 | 29.45 | 46.75 |
| 10 | AG-Residual | Def | IDW | 30s | 63.76 | 92.84 | 59.17 | 60.14 | 37.57 | 48.20 | 54.50 | 29.53 | 46.78 |
| 11 | AG-Residual | Def | IDW | 10s | 63.80 | 92.81 | 59.60 | 60.38 | 38.87 | 48.56 | 54.86 | 29.62 | 46.83 |

Table 4: Ablation studies on MovieNet-SSeg, BBC, and MovieChat-SSeg datasets. We evaluate genre embedding integration methods (rows 1-3, 8), prompt strategies (rows 4-5, 8), anchor sampling approaches (rows 6-7, 8), and test-time shot splitting thresholds (rows 8-11).

affinity-based residual method. Frame-level concatenation selects the most similar genre embedding for each shot and concatenates it with the shot-level representation, while token-level concatenation adds genre embeddings to the image token sequence before passing through ViT attention blocks. As shown in Tab. 4(rows 1-3, 8), frame-level concatenation actually degrades performance compared to using no genre information (59.13 vs 61.27 AP), while token-level concatenation provides modest improvements. Our affinity-based residual approach achieves the best performance (63.62 AP), demonstrating that dynamic integration of genre information through attention mechanisms is more effective than simple concatenation strategies. These results indicate that the method of incorporating genre metadata significantly impacts its effectiveness for scene segmentation.

**Genre Prompt Design** We evaluate three genre prompt strategies to generate genre embeddings: name-based (using only genre names), style-based (describing visual and narrative characteristics), and definition-based (incorporating comprehensive IMDb definitions). As shown in Tab. 4(rows 4-5, 8), definition-based prompts achieve the highest performance (63.62 AP), outperforming name-based approaches by 0.39 points. This suggests that more detailed textual descriptions provide better semantic guidance for genre-aware representation learning. The complete list of prompts used for each genre is provided in the appendix.

**Shot Duration** We evaluate three anchor sampling strategies for pseudo-boundary generation: side-based selection using fixed endpoints, duration-weighted(DW) sampling that prioritizes longer shots, and our inverse-duration-weighted(IDW) approach that favors shorter shots. As shown in Tab. 4(rows 6-7, 8), our inverse-duration-weighted strategy achieves the best performance, improving AP by 0.8 over the side-based baseline and 0.18 over duration-weighted sampling. This result demonstrates that prioritizing shorter shots as anchors provides more diverse training signals and enhances representation learning compared to conventional or length-biased approaches.

**Shot Split Threshold** We evaluate the effectiveness of test-time shot splitting with different duration thresholds to address challenges from long shots during inference. As shown in Tab. 4(rows 8-11), progressively smaller thresholds lead to consistent performance improvements: no splitting (63.62 AP) $\rightarrow$ 60s threshold (63.63 AP) $\rightarrow$ 30s threshold (63.76 AP) $\rightarrow$ 10s threshold (63.80 AP). The 10-second threshold achieves the best performance, suggesting that subdividing long shots into smaller segments improves boundary detection accuracy.

## 5.6 QUALITATIVE ANALYSIS OF FAILURE CASES

We analyze the three lowest-performing videos in MovieNet-SSeg test set to identify systematic failure patterns. Fig. 3 illustrates representative cases across five common error types.

**Insert shots and non-narrative elements.** Brief inserts such as television screens, photographs, or archival footage are frequently misclassified as scene boundaries. Our visual-only approach lacks narrative understanding to recognize these storytelling devices maintain continuity with surrounding sequences.

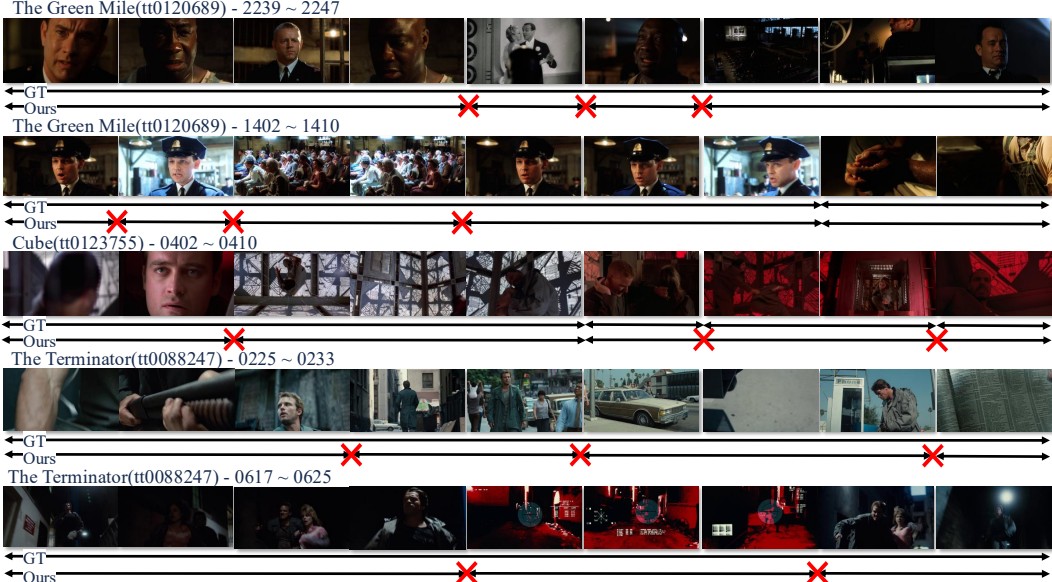

Figure 3: Common failure patterns in scene segmentation. Ground truth (GT) boundaries are shown with black arrows; our predictions with red X marks indicating errors.

**Lighting variations.** Sudden lighting shifts within continuous scenes trigger false boundaries. Lightning flashes, flickering lights, or rapid shadow changes create visual discontinuities misinterpreted as scene changes, revealing over-reliance on visual continuity cues.

**Subtle spatial transitions.** Gradual location shifts while maintaining narrative flow are often missed. When scenes expand from interior to exterior or between adjacent locations, high visual similarity leads to under-segmentation, indicating limited semantic understanding of spatial context.

**Extreme framing and unconventional compositions.** Extreme close-ups or unusual camera angles create distinctive features that sometimes trigger spurious boundaries, revealing sensitivity to out-of-distribution visual patterns.

**Special effects and visual artifacts.** Digital effects such as compositing or CGI create scene-irrelevant visual discontinuities misinterpreted as boundaries, producing strong visual cues that mislead our appearance-based approach.

## 6 CONCLUSION

In this paper, we present a novel approach for video scene segmentation that leverages production-level metadata through genre-guided representation learning, duration-aware anchor sampling, and test-time shot splitting. We introduce MovieChat-SSeg, a scene segmentation benchmark comprising 1,000 human-annotated video clips across diverse content types, supporting evaluation across varied narrative structures. Extensive experiments demonstrate that our method achieves state-of-the-art performance on multiple benchmarks, with significant improvements over existing approaches across diverse video content types.

**Limitations.** While our method successfully incorporates genre and temporal metadata, it may face challenges in content where visual and production cues are insufficient for scene boundary detection, such as dialogue-heavy scenes with minimal visual variation. Additionally, our approach focuses on movie-style content and may require adaptation for other video formats with different production conventions. Our current pretraining relies solely on MovieNet data, and integrating diverse large-scale video datasets could potentially improve generalization across varied content types. Furthermore, evaluation on broader video categories (e.g., news broadcasts, vlogs, educational content) would provide insights into the method's applicability beyond narrative-driven content. Future work could explore additional production metadata and multi-modal approaches Argaw et al. (2023) incorporating audio and dialogue information for more comprehensive scene understanding.

## ACKNOWLEDGEMENTS

We thank Dr. ChiHoon Lee, Chief AI Officer of the AI R&D Division at CJ Corporation, and colleagues at CJ Corporation, for their insightful discussions and guidance that helped shape the direction of this work. This work was supported by the National Research Foundation of Korea(NRF) grant funded by the Korea government(MSIT)(RS-2024-00338439) and Institute of Information & Communications Technology Planning & Evaluation (IITP) grants funded by the Korea government (MSIT) (RS-2020-II201361).

## REPRODUCIBILITY STATEMENT

To ensure reproducibility, we provide comprehensive implementation details and resources. Our supplementary materials include source code and sample data from our newly created MovieChat-SSeg dataset. Detailed pseudocode for key algorithmic components—including the genre embedding integration process and duration-aware anchor sampling for pseudo-boundary generation—is presented in Appendix I. Implementation details, including hyperparameters, training procedures, and architectural specifications, are provided in Section 5.3. Upon acceptance, we will publicly release our complete codebase, dataset processing scripts, and the full MovieChat-SSeg dataset to facilitate future research and ensure full reproducibility of our results.

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

## A    APPENDIX

Our supplementary material provides comprehensive details complementing the main paper. It covers the MovieChat-SSeg dataset, including annotation procedures, quality assurance, and statistical analyses in Section B. We validate genre embeddings through zero-shot classification, explore prompt design strategies, analyze embedding similarities across different vision-language models, examine various types of production metadata and conduct clustering analysis to further validate our learned representations in Section C. Additionally, we assess robustness through genre vocabulary reduction experiments (21→6 genres) and analyze performance trends across videos with 1-5 genres per video in SectionC. Section D discusses pseudo-boundary search techniques within shot sequences and validates our duration-aware anchor sampling approach. We analyze the computational overhead of test-time shot splitting, evaluate performance-efficiency trade-offs across different duration thresholds, and demonstrate the practical feasibility of our approach for deployment in Section E. We provide comparisons with recent supervised methods in Section F. Finally, additional qualitative results are presented for the BBC dataset, generalization analysis on non-narrative formats (lectures, gaming) from the LongerVideos dataset and a practical application on the BBDB baseball dataset in Section H.

## B    MOVIECHAT-SSEG DATASET

The MovieChat-SSeg dataset extends the MovieChat-1K (Song et al. (2024)) dataset by adding scene boundary annotations to 1K videos. These videos are trimmed to about 8 minutes each and include a variety of content types, such as movies, TV shows, and documentaries. A scene consists of one or more shots, and the annotations are manually created following the shot-based approach used in MovieNet (Huang et al. (2020)).

**Annotation Process**    The videos are divided into shots using shot boundary detection (Castellano (2014)). Within each shot, keyframes are extracted through uniform sampling, and we select 3 keyframes. As shown in Fig. 4, we develop a web-based annotation tool. Annotators view the shot sequence composed of keyframes, as shown in Fig. 4(a), and determine whether the target shot is a scene boundary. To provide contextual information, surrounding shots are also displayed. Annotators are given the option to view 3, 5, or 7 shots at once, with most annotators using 5 consecutive shots. The timeline at the bottom of the tool allows annotators to use shot duration, which can be helpful for ambiguous cases, such as long takes, where keyframe-based judgment alone might be insufficient. Annotators can switch to video mode and review the actual video, as shown in Fig. 4(b).

**Annotation Quality**    Annotators receive training before beginning their annotation tasks. This training includes hands-on practice with the MovieNet-SSeg dataset, where annotators perform annotations and compare their results with the ground truth (GT). This process helps them better understand the concept of scene boundaries. To ensure high-quality annotations, we employ a multi-annotator process: each video is independently annotated by two, and any disagreements are resolved by a third annotator. On average, annotation takes approximately 2 minutes per video by each annotator. The average agreement rate between the first two annotators is 38%. The third annotator reviews the video and makes corrections based on the annotations provided by the first two.

**Statistics**    As shown in Tab. 5, the dataset contains diverse video formats exhibiting distinct shot and scene characteristics. For example, documentaries tend to have longer shots and scenes, whereas TV shows have more shots and fewer scene transitions compared to movies. Unlike prior datasets Huang et al. (2020); Rotman et al. (2016), MovieChat-SSeg enables robust evaluation across varied narrative styles and editing conventions. By linking videos to IMDb-based genre tags, we confirm broad genre coverage. Combined with its diverse shot and scene structures, MovieChat-SSeg serves as a valuable benchmark for developing generalizable scene segmentation methods across domains.

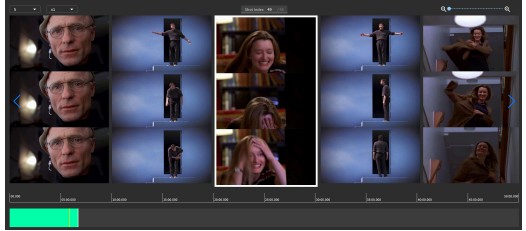 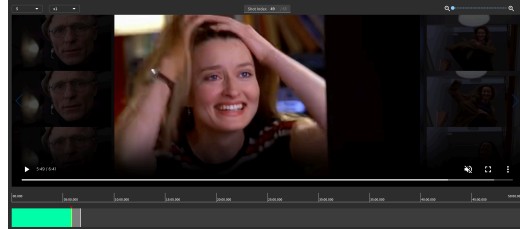

(a) Shot Mode                                                      (a) Video Mode

Figure 4: Annotation interface. In Shot Mode, annotators assess scene boundaries based on keyframe sequences, while in Video Mode, they can review the actual video to refine their annotation decisions.

| Type | #Videos | Avg Duration(s) | Avg Shots | Avg Scenes | Genre Tags |
|------|---------|-----------------|-----------|------------|------------|
| **Movie** | 531 | 460.15 | 119.47 | 8.98 | 20(Except Documentary) |
| **TV Show** | 252 | 421.41 | 132.83 | 7.15 | 8(Act, Adv, Cri, Dra, Fan, Mys, Rom, Thr) |
| **Documentary** | 217 | 448.48 | 90.00 | 5.81 | 5(Adv, Ani, Doc, Fam, His) |
| **Total** | 1000 | 447.86 | 116.44 | 7.83 | 21 |

Table 5: Statistics of shot and scene counts for different video types in MovieChat-SSeg.

## C  GENRE EMBEDDING

**Genre Embedding Validation**  To verify the relevance of genre information to scene-level visual representations, we conduct a zero-shot classification experiment on the MovieNet-SSeg dataset. First, we obtain shot-level visual features by passing keyframes through the OpenCLIP Cherti et al. (2023) image encoder and averaging the resulting embeddings. Scene-level representations are then computed by averaging shot embeddings within each scene. Genre vectors are obtained by encoding IMDb-based textual genre prompts using the OpenCLIP text encoder. Using these scene-level visual features and genre embeddings, we perform zero-shot classification via cosine similarity. For evaluation, each movie has $n$ genre tags, and we measure the top-$n$ accuracy by checking whether the true genres are among the top-$n$ predictions. Our method achieves a top-$n$ accuracy of 60.52%. The zero-shot classification procedure is summarized in Algorithm 1. Additionally, t-SNE visualization of the scene-level features reveals distinct clusters aligned with genre categories, supporting the hypothesis that genre metadata—although annotated at the movie level—provides meaningful semantic guidance at the scene level (see Fig. 5). Based on these insights, we incorporate genre information into shot representation learning to enhance video scene segmentation performance.

---
**Algorithm 1** Zero-shot Scene-Level Genre Classification
---
1: **Input:** Keyframes of shots $\{k_j\}$, genre prompts $\{g_m\}$
2: Compute shot representations $e_i$ by averaging CLIP image embeddings of $k_j$
3: Compute scene representations $s_s$ by averaging $e_i$ within each scene
4: Compute genre embeddings $g_m$ from textual prompts via CLIP text encoder
5: **for** each scene $s_s$ **do**
6:    Compute cosine similarity between $s_s$ and each $g_m$
7:    Select top-$n$ genres by similarity
8:    Evaluate if true genre tags are in top-$n$
9: **end for**
10: **Output:** Top-$n$ accuracy
---

**Genre Prompt Design**  We categorize genre conventions into three types: name, style, and definition. Inspired by CLIP Radford et al. (2021)'s prompts for zero-shot classification, we construct prompts according to types for each genre. First, we simply formulate the name type as '[genre name], a film genre' (e.g., 'Drama, a film genre'). In the style category, prompts describe common visual characteristics of each genre by extending the name format with additional descriptions shown in Tab. 17. For the definition category, we incorporate IMDB's genre tagging guidelines IMDb.com (2023), extending the name format with additional prompts shown in Tab. 18.

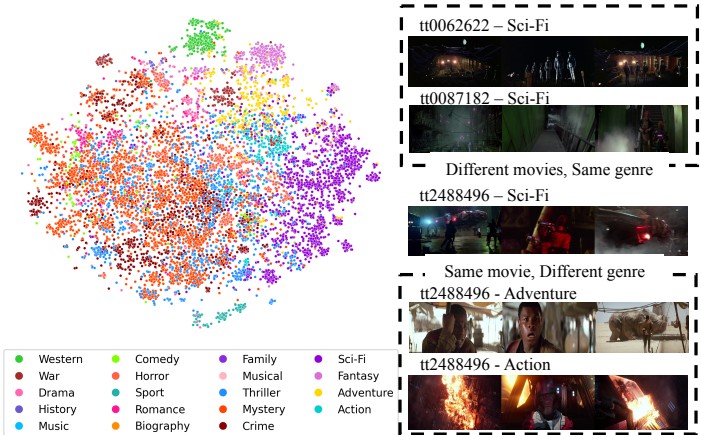

Figure 5: Each point represents a scene's visual representation obtained by averaging shot-level features extracted from CLIP image encoder. The emergence of genre-specific clusters suggests that scenes within the same genre share common visual characteristics, despite the video-level genre annotations.

**Genre Embedding Analysis**  We use the embedding vectors obtained by inputting these prompts into OpenCLIP Cherti et al. (2023)'s text encoder as representations that encapsulate each genre. We analyze relationships among film genres in Fig. 6 by comparing embeddings from definition prompts (y-axis) against those from name, style, and definition prompts (x-axis). These maps show high similarities along their diagonals, indicating that each genre maintains its distinctive characteristics across name, style, and definition prompts. As shown in Fig. 6(c), the figure demonstrates that definition embeddings effectively distinguish between genres, while capturing natural relationships between similar genre pairs.

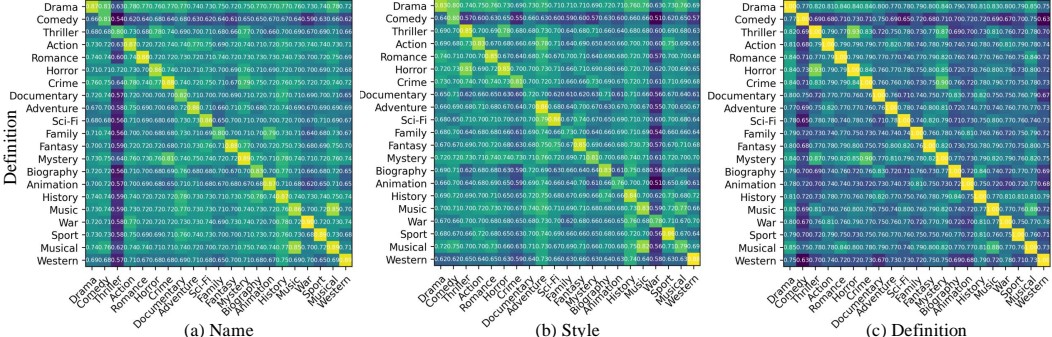

Figure 6: Similarity analysis of genre embeddings across different prompt types (name, style, and definition), where brighter colors indicate higher similarities. With definition-based prompts fixed on the y-axis, the consistent patterns of genre embedding similarities across different prompt types suggest that genre-specific characteristics are effectively captured regardless of prompt variations.

**Different Vision-Language Models**  Our approach offers flexibility through component-wise substitution, allowing different vision-language (V-L) text encoders to be integrated for genre embedding. As shown in Tab. 6, consistent performance improvements are observed across various V-L text encoders, including BLIP-2 Li et al. (2023) and LanguageBind Zhu et al. (2023). This demonstrates that the text encoders robustly capture genre characteristics expressed through our prompt-based framework. These findings validate the generality and adaptability of our method to different pretrained language models, highlighting its potential applicability as new and more powerful V-L models continue to emerge. Furthermore, consistent gains indicate that our genre embedding strategy is not dependent on a specific encoder architecture, underscoring its robustness and flexibility.

**Integration of Various Metadata**  We further evaluate the effectiveness of incorporating different types of metadata into shot representation learning, as presented in Tab. 7.

| Method | AP | AUC | F1 | mIoU |
|---|---|---|---|---|
| BLIP-2 Li et al. (2023) | 63.44 | 92.73 | **59.41** | **59.85** |
| LanguageBind Zhu et al. (2023) | 63.31 | 92.79 | 58.82 | 59.53 |
| Ours | **63.62** | **92.84** | 58.88 | 59.64 |

Table 6: Vision-Language Models for Genre Embedding

Specifically, we examine shot scale, shot angle, camera movement, and color grading —all metadata linked to visual storytelling techniques influenced by directorial intent during production. Similar to our approach for genre, we construct textual prompts based on category-specific definitions within each metadata type and embed these prompts to generate semantic vectors. Our results confirm that integrating such metadata within our framework yields consistent performance improvements over approaches that exclude metadata information. These encouraging outcomes suggest that leveraging a wider range of metadata types could further enhance the robustness and accuracy of video scene segmentation in future work.

| Method | AP | AUC | F1 | mIoU |
|---|---|---|---|---|
| Shot Scale | 63.58 | 92.72 | 58.82 | 59.51 |
| Shot Angle | 63.40 | 92.75 | 58.72 | 59.57 |
| Camera Movement | 63.24 | 92.71 | 58.82 | 59.46 |
| Color Grading | 63.57 | 92.83 | 58.78 | 59.43 |
| Ours | **63.62** | **92.84** | **58.88** | **59.64** |

Table 7: Different types of Metadata

**Clustering Analysis** To further validate our shot representation learning, we conduct unsupervised clustering experiments on 64 movies from the MovieNet-SSeg test set. Using learned shot-level features, we perform hierarchical clustering with the number of clusters set equal to the ground truth number of scenes per movie. The results show that genre integration improves clustering performance significantly (+0.088 ARI, +0.127 AMI), demonstrating that our genre-guided features capture meaningful semantic relationships for scene organization, as shown in Tab. 8 These clustering results provide complementary evidence to our boundary detection experiments, showing that the learned representations naturally group shots into semantically coherent clusters even without explicit boundary supervision during clustering.

| Method | ARI | AMI |
|---|---|---|
| BaSSL | 0.209 | 0.486 |
| TranS4mer | 0.257 | 0.544 |
| Ours (w/o genre) | 0.146 | 0.390 |
| Ours (w/ genre) | 0.234 | 0.517 |

Table 8: Clustering performance using Adjusted Rand Index (ARI) and Adjusted Mutual Information (AMI). Higher values indicate better agreement with ground truth scene boundaries.

**Genre Vocabulary Robustness** We evaluate robustness to reduced genre coverage by progressively removing the least frequent genres from the embedding pool (21, 16, 11, 6 genres). Tab. 9 shows non-linear performance degradation, with AP highest at 21 genres, followed by 16, then 6, and lowest at 11 genres. The 6-genre configuration unexpectedly outperforms 11 genres, suggesting that removing additional niche genres eliminates conflicting priors, allowing the model to focus on core genre patterns. Critically, even with only 6 genres (approximately 30% of vocabulary), our method outperforms the metadata-free baseline by 2.14 AP, confirming that genre priors provide value regardless of coverage completeness. This demonstrates practical applicability to domains with limited genre taxonomies.

**Impact of Genre Count on Performance** Our method naturally handles multi-genre videos without explicit modifications, as genre embeddings integrate through affinity mapping with visual to-

| #Genre | AP | AUC | F1 | mIoU |
|---|---|---|---|---|
| 21 | 63.62 | 92.84 | 58.88 | 59.64 |
| 16 | 63.54 | 92.72 | 59.14 | 59.71 |
| 11 | 63.33 | 92.67 | 58.93 | 59.71 |
| 6 | 63.41 | 92.75 | 58.96 | 59.60 |

Table 9: Robustness to reduced genre coverage. Least frequent genres are progressively removed from the embedding pool.

kens during shot representation learning rather than shot-level prediction. We analyze performance across videos with varying genre counts in MovieNet-SSeg test set as shown in Tab. 10. Results show monotonic performance degradation as genre count increases, with AP declining from 67.20 (1 genre) to 62.38 (4 genres). This trend reflects two factors. First, multi-genre videos exhibit complex narrative structures where multiple storytelling conventions interweave, creating ambiguous scene boundaries that do not align cleanly with single-genre patterns. Second, as genre count increases, individual genre priors provide less distinctive guidance as our affinity-based integration must balance competing conventions. Despite this challenge, our soft integration mechanism accommodates multi-genre complexity by dynamically weighting relevant priors.

| #Genre | #Title | AP | AUC | F1 | mIoU |
|---|---|---|---|---|---|
| 1 | 3 | 67.20 | 93.29 | 59.11 | 58.13 |
| 2 | 18 | 65.69 | 93.76 | 60.50 | 59.33 |
| 3 | 23 | 63.42 | 92.23 | 58.20 | 59.72 |
| 4 | 19 | 62.38 | 93.30 | 59.13 | 60.10 |
| 5 | 1 | 58.66 | 87.48 | 51.95 | 55.33 |

Table 10: Impact of genre count on performance. #Title indicates the number of movies in each category.

## D    PSEUDO-BOUNDARY SEARCH

**Scene Boundaries in a Shot Sequence** In video scene segmentation, a sliding window approach Rao et al. (2020a); Chen et al. (2021); Mun et al. (2022); Islam et al. (2023); Yang et al. (2023); Chen et al. (2024) is commonly used to capture contextual information from surrounding shots. Existing approaches configure their sequence length between 17 Mun et al. (2022) to 25 Islam et al. (2023) shots as a hyper-parameter to achieve optimal benchmark performance. Following BaSSL's protocol Mun et al. (2022), we use a sequence of 17 shots with 8 neighboring shots on each side. During pre-training, we detect pseudo boundaries within shot sequences to learn shot representations and inter-shot relations without scene boundary annotations. These pseudo boundaries must effectively reflect the characteristics of actual scene boundaries. As shown in Fig. 7(a), since many shot sequences contain multiple scene boundaries, allowing multiple pseudo boundaries within a shot sequence proves beneficial for learning shot representations.

**Pseudo Boundaries in a Shot Sequence** We search for pseudo-boundaries by sampling anchors based on shot duration and applying Dynamic Time Warping (DTW) Berndt & Clifford (1994) with representation similarity between anchors and the shot sequence. The sampling weights are inversely proportional to shot duration, favoring shorter shots. As shown in Fig. 7(b), we demonstrate the number of unique anchor combinations obtained through sampling performed 20 times, matching the number of pretraining epochs. Although diverse anchor combinations are formed, we observe that the number of pseudo-boundaries converges to a meaningful subset of potential boundaries, demonstrating its effectiveness with respect to shot representation learning, as shown in Fig. 7(c). Additionally, as illustrated in the following Fig. 8, we show the examples how various combinations of anchors generate diverse pseudo boundaries.

**Pseudo boundary validation against ground truth annotations** We perform comparative experiments between fixed anchor and our proposed method to assess the similarity between generated pseudo boundaries and ground truth scene boundaries, obtaining the results presented in Tab. 11.

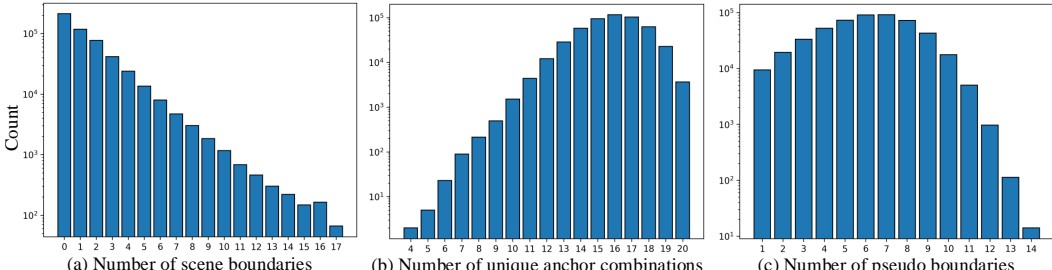

Figure 7: Distribution analysis of boundaries in shot sequences: (a) number of actual scene boundaries per 17-shot sequence in MovieNet-SSeg dataset, showing significant cases of multiple boundaries, (b) diversity of anchor combinations obtained through shot duration-based sampling, and (c) resulting pseudo boundaries generated from these anchor combinations, demonstrating our model's capability to capture multiple scene transitions within a sequence.

While fixed anchors generate identical pseudo boundaries for each shot sequence, our probabilistic approach produces diverse results across iterations. To evaluate boundary discovery effectiveness, we measured coverage—the proportion of actual boundaries successfully identified. For sequences with multiple scene boundaries, fixed anchors can match at most one boundary due to deterministic anchor selection, whereas our probabilistic sampling generates varied pseudo boundaries across training iterations, increasing alignment probability with actual boundaries. This diversity enhances shot representation learning by exposing the model to multiple realistic transition patterns, enabling more robust and generalizable embeddings for scene segmentation.

| Sequence Complexity | #Samples | #Boundaries | Fixed Anchor | Probabilistic Anchor | Improvement |
|---|---|---|---|---|---|
| 0 boundary | 45,488 | 0 | - | - | - |
| 1 boundary | 25,240 | 25,240 | 0.285 | 0.463 | +0.178 |
| 2 boundary | 16,396 | 32,792 | 0.177 | 0.411 | +0.234 |
| 3 boundary | 8,560 | 25,680 | 0.137 | 0.391 | +0.254 |
| 4 boundary | 4,565 | 18,260 | 0.107 | 0.371 | +0.264 |

Table 11: Boundary coverage comparison between fixed anchor and probabilistic (duration-aware) anchor sampling across different sequence complexities.

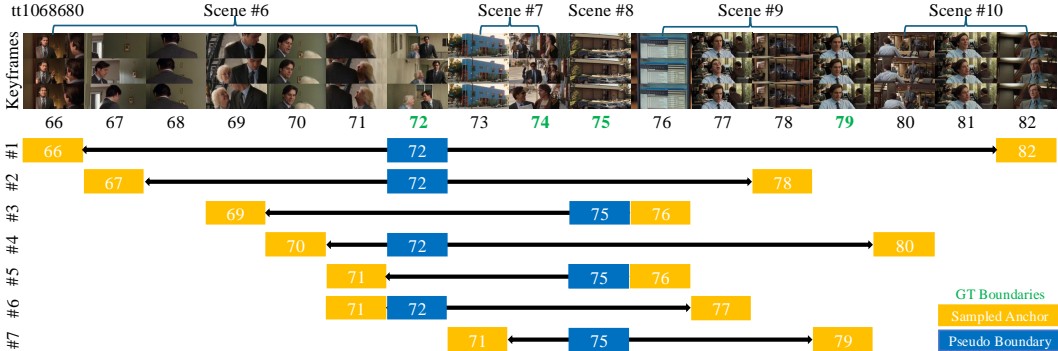

Figure 8: Visualization of pseudo boundary search process with diverse anchor combinations in a shot sequence. Multiple ground truth scene boundaries (in green) exist within the sequence, and our approach generates pseudo boundaries (in blue) through various anchor combinations (in yellow). The example demonstrates how our anchor samplings lead to pseudo boundaries that effectively align with actual scene transitions, facilitating better shot representation learning during pretraining.

## E    TEST-TIME SHOT SPLITTING

**Computational Cost of Test-Time Shot Splitting** Test-time splitting introduces minimal computational overhead, as additional processing is required only for shots exceeding the duration threshold. Tab. 12 shows the number of split shots and resulting total shot counts in MovieNet-SSeg test set

across different thresholds. Since our model processes shots independently, computational cost scales linearly with shot count. At our recommended 10-second threshold, 8,811 shots are split (8.3% of total), increasing the shot count from 105,976 to 123,598. This represents approximately 16.6% computational overhead—a favorable trade-off given the 0.18 AP improvement over the no-split baseline. Higher thresholds (30s, 60s) further reduce overhead to 2.0% and 0.4% respectively, though with diminished performance gains.

| Threshold | Split Shots | Total Shots | Overhead |
|---|---|---|---|
| None (baseline) | 0 | 105,976 | - |
| 60s | 210 | 106,396 | +0.4% |
| 30s | 1,046 | 108,068 | +2.0% |
| 10s | 8,811 | 123,598 | +16.6% |

Table 12: Computational overhead of test-time shot splitting at different thresholds in MovieNet-SSeg test set.

## F    COMPARISON WITH RECENT METHODS

We compare our approach with recent supervised scene segmentation methods that focus on contextual learning from shot-level features:

| Method | Year | AP | AUC | F1 | mIoU |
|---|---|---|---|---|---|
| LGSS | 2020 | 47.10 | - | - | 48.80 |
| VSMBD* Tan et al. (2024b) | 2024 | 49.17 | 87.16 | 35.87 | 43.47 |
| NeighborNet* Tan et al. (2024a) | 2024 | 50.07 | 87.54 | 40.89 | 47.76 |
| MASRC* Tan et al. (2025) | 2025 | 52.62 | 89.39 | 45.84 | 49.63 |
| Ours | - | 63.62 | 92.84 | 58.88 | 59.64 |

Table 13: Comparison with recent supervised methods on MovieNet-SSeg. *Methods reproduced using available implementations with best-effort parameter tuning.

Note that our reproduced results may not fully match the originally reported performance due to potential differences in implementation details, hyperparameters, or evaluation protocols. These methods primarily focus on supervised contextual learning, while our approach emphasizes self-supervised representation learning with genre guidance.

## G    DIALOGUE AND GENRE-SPECIFIC ANALYSIS

**Performance across dialogue levels** We analyze our method's performance across different dialogue densities to understand the impact of visual-only features in Tab. 14. We align dialogue to shots by matching timestamps in subtitles with shot boundaries in the MovieNet-SSeg test set. Dialogue density is computed as total subtitle characters divided by average shot duration per scene, then categorized into tertiles (Low/Medium/High). Results show that AP decreases from 76.92 (Low) to 61.62 (High) as dialogue density increases, confirming that visual features alone struggle with dialogue-driven scenes.

| Dialogue Density | #Scene | AP | AUC | F1 | mIoU |
|---|---|---|---|---|---|
| Low | 2676 | 76.92 | 87.80 | 60.00 | 47.23 |
| Medium | 2758 | 70.73 | 89.69 | 60.20 | 50.72 |
| High | 2676 | 61.62 | 95.20 | 57.25 | 58.48 |

Table 14: Performance by dialogue density in MovieNet-SSeg.

**Genre-specific test-time split analysis** We further analyze test-time split effectiveness across genres in Tab. 15. The MovieNet-SSeg test set contains 64 movies spanning 17 genres (avg. 2.95 genres

per movie). Test-time shot split shows genre-dependent effectiveness. Mystery/Crime genres benefit substantially, as their longer shots can be subdivided into finer boundaries, improving precision in detecting narrative transitions. Conversely, Adventure/War genres show limited or negative impact because their rapid action sequences already produce short shots that offer minimal room for meaningful subdivision. This demonstrates our method's adaptive behavior, where splitting enhances segmentation for extended compositions while avoiding over-segmentation in fast-paced content.

| Genre | #Title | w/o Split | w/ Split | Improvment |
|---|---|---|---|---|
| Drama | 37 | 66.25 | 66.44 | 0.19 |
| Thriller | 26 | 59.91 | 60.13 | 0.22 |
| Action | 20 | 59.94 | 60.09 | 0.15 |
| Crime | 16 | 60.81 | 61.37 | 0.56 |
| Romance | 16 | 71.33 | 71.53 | 0.20 |
| Adventure | 15 | 61.76 | 61.59 | -0.17 |
| Sci-Fi | 14 | 58.38 | 58.66 | 0.28 |
| Mystery | 10 | 57.64 | 58.17 | 0.53 |
| Comedy | 9 | 67.30 | 67.48 | 0.18 |
| Fantasy | 7 | 61.86 | 62.07 | 0.21 |
| War | 6 | 60.87 | 60.41 | -0.46 |
| Biography | 4 | 74.52 | 74.58 | 0.06 |
| History | 4 | 71.04 | 70.55 | -0.49 |
| Music | 2 | 66.48 | 67.26 | 0.78 |
| Horror | 1 | 69.78 | 71.46 | 1.68 |
| Musical | 1 | 58.66 | 58.32 | -0.34 |
| Family | 1 | 68.52 | 67.79 | -0.73 |

Table 15: Genre-Specific Impact of Test-Time Shot Split. Performance is measured by AP (Average Precision). Test-time split threshold = 10 seconds

## H    ADDITIONAL QUALITATIVE RESULTS

**BBC dataset** The results on BBC Baraldi et al. (2015) dataset demonstrate our model's generalization capability to TV series, where we present examples of accurately detected scene transitions in various video formats. As shown in Fig. 9, existing approaches like BaSSL Mun et al. (2022) and TranS4mer Islam et al. (2023) exhibit limitations in handling significant visual variations within scenes. In contrast, our method validate the robust performance by maintaining the consistency in scene boundary detection across diverse environmental conditions. This performance shows our method's ability to capture high-level semantic changes while being resilient to low-level visual variations, making it particularly suitable for professional broadcast contents.

**Generalization to non-narrative video formats** We explore our method's applicability to non-narrative formats using the LongerVideos Ren et al. (2025) dataset, including lectures and entertainment content (gaming, award ceremonies). Qualitative analysis reveals fundamental limitations as shown in Fig. 10. Lecture videos show poor shot detection, resulting in overly sparse segmentation compared to video length. Entertainment content similarly fails to identify boundaries adequately. These failures highlight that "scene" definitions in non-narrative content fundamentally differ from narrative structure. Lectures are organized by topic transitions rather than spatial-temporal changes, while gaming videos and award shows follow event-based structures that diverge from cinematic conventions. Additionally, these formats rely heavily on dialogue and audio for context, which our visual-only approach cannot capture.

**Application to BBDB dataset** To demonstrate the applicability of our method for general video segmentation, we apply it to the Baseball Database (BBDB) Shim et al. (2018) dataset - a collection of baseball game broadcasts that differs substantially from our training data. First, we apply shot boundary detection Castellano (2014) to segment videos into shots and extract keyframes. These keyframes are then fed into our method to detect scene boundaries. After detecting scene boundaries, we merge shots within the same scenes. Finally, we filter these merged scenes based on scene duration to obtain actual baseball game segments. By identifying and removing brief scenes, we can

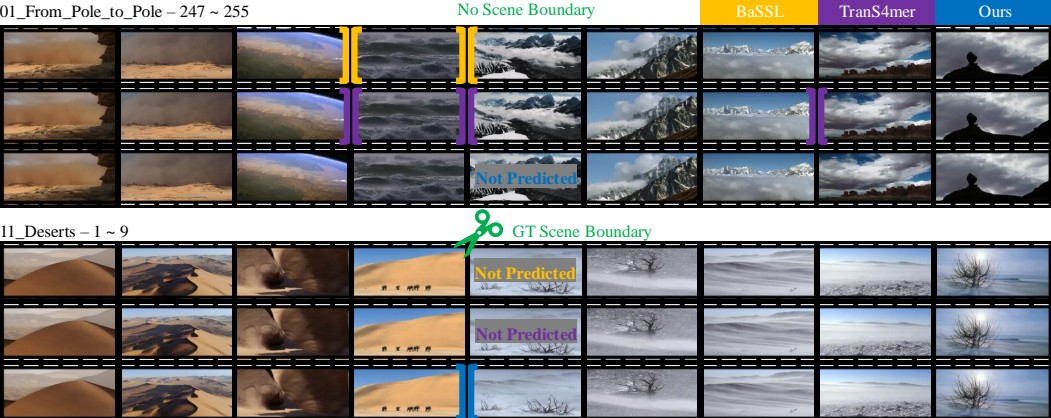

Figure 9: Qualitative comparison of scene boundary detection methods on BBC Planet Earth sequences. Top: "From Pole to Pole" featuring polar landscapes. Bottom: "Deserts" showing varied terrain. Orange/purple boxes indicate false positives/negatives from baseline methods, while our approach (blue) correctly identifies scene boundaries aligned with ground truth (green)

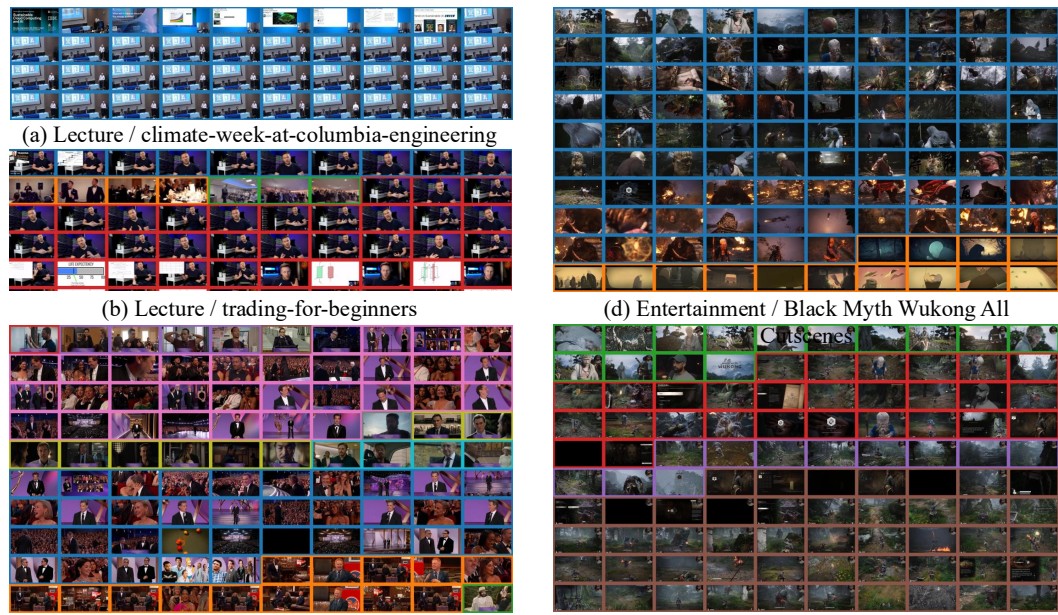

(a) Lecture / climate-week-at-columbia-engineering

(b) Lecture / trading-for-beginners

(d) Entertainment / Black Myth Wukong All

(c) Entertainment / 76th Primetime Emmy Awards

(e) Entertainment / Black Myth Wukong - Part 1

Figure 10: Qualitative Results on Non-Narrative Content. Our method shows fundamental limitations on lecture and entertainment videos, where scene definitions differ from narrative structure. Shots in the same scene share the same border color.

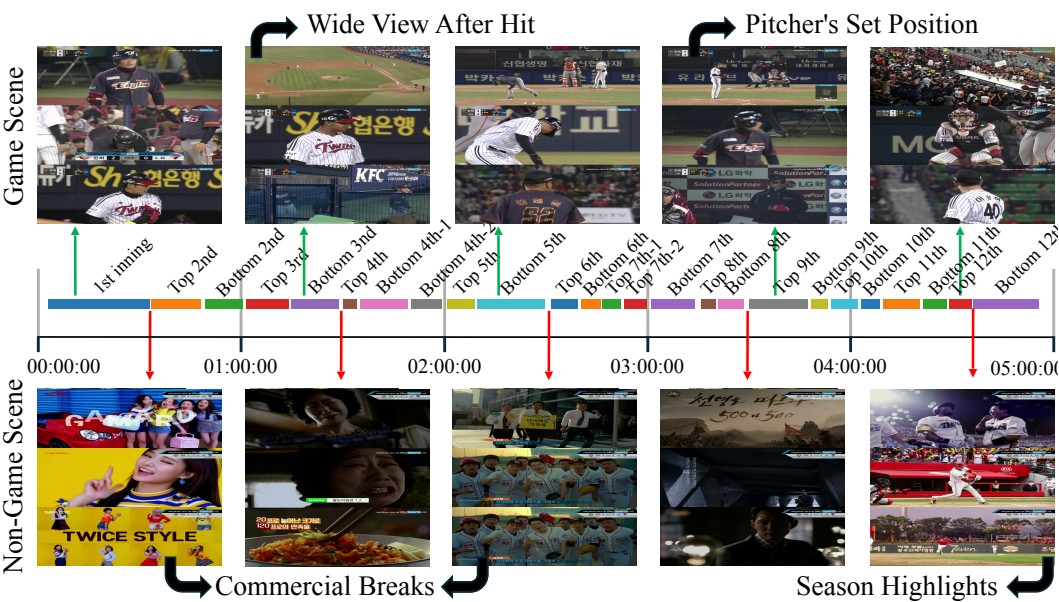

Figure 11: Timeline visualization of scene segments in a baseball game video from BBDB.

focus on significant game segments. These longer segments typically represent key transitions in the game, such as changes between innings or pitcher replacements, which are marked by distinct scene changes, as shown in Fig. 11. We visualize scene segmentation results on BBDB dataset to demonstrate practical utility, showing how our model effectively organizes video content into coherent scenes. As shown in Fig. 12, our method successfully distinguishes between game-related scenes containing actual gameplay and non-game content such as advertisements between gameplay segments.

We present a simple application leveraging scene segmentation on the BBDB baseball broadcast dataset. By filtering scenes based on their duration, we obtain segments of meaningful lengths(over 180 seconds) that better correspond to actual game events. As shown in Table 16, the full dataset consists of 1,172 videos with a total runtime of approximately 4,236 hours and over 237,000 scenes. After filtering, this is reduced to 23,528 scenes spanning about 3,588 hours. This corresponds to an average of around 20 scenes per game, reflecting a semantic segmentation aligned with the baseball game structure. Importantly, the filtered scenes cover 94.75% of the game event annotations, despite representing only 84.69% of the total video duration. This suggests that scene-based filtering effectively narrows down the temporal search space while retaining most of the key events. Such segmentation can thus facilitate downstream tasks like temporal localization and highlight generation by reducing search complexity and enabling more focused analysis on semantically coherent video segments.

| Type | #Videos | #Shots | #Scenes | Duration(H) | #Plays |
|------|---------|--------|---------|-------------|--------|
| **Total** | 1172 | 2,917,186 | 237,913 | 4236.87 | 404,964 |
| **Filtered** | 1172 | 1,708,972 | 23,528 | 3588.34 | 383,711 |

Table 16: Statistics of the BBDB baseball broadcast dataset before and after scene duration-based filtering. Filtering reduces the total duration and number of scenes while retaining the majority of annotated game events, demonstrating efficient temporal segmentation.

# I   ALGORITHM FLOW

We provide our ViT-based shot encoder incorporating our approach in pseudocode form to explain how genre embedding operates and integrates with visual features.

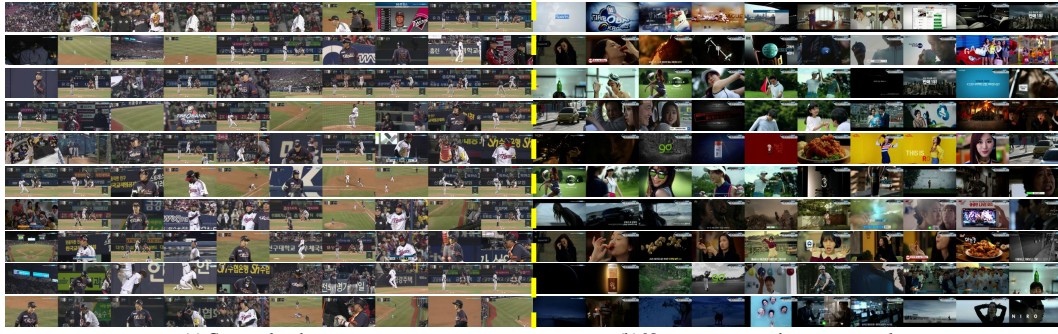

(a) Game-related scenes          (b) Non-game content between gameplay segments

Figure 12: Visualization of segmented scenes from BBDB dataset. (a) Baseball game-related scenes showing actual gameplay and related activities. (b) Non-game scenes including advertisements and miscellaneous content between gameplay segments.

---

**Algorithm 2** Genre-Aware Visual Feature Integration

---

1: **Input:** Visual tokens $V_t \in \mathbb{R}^{N_t \times D}$
2: **Model parameters:** Genre embeddings $G_e \in \mathbb{R}^{N_g \times D}$, Projection matrix $W \in \mathbb{R}^{D \times D}$
3:
4: // Compute genre-visual affinity matrix
5: **for** $i = 1$ **to** $N_t$ **do**
6:      **for** $j = 1$ **to** $N_g$ **do**
7:          affinity$[i, j] \leftarrow \frac{V_t[i] \cdot G_e[j]}{\sqrt{D}}$
8:      **end for**
9: **end for**
10: $A \leftarrow \text{softmax}(\text{affinity})$ // normalize affinity scores
11:
12: // Project genre embeddings
13: $G_{\text{projected}} \leftarrow []$
14: **for** $j = 1$ **to** $N_g$ **do**
15:      $g_{\text{proj}} \leftarrow G_e[j] \times W$ // apply learnable projection
16:      $G_{\text{projected}} \leftarrow \{G_{\text{projected}}; g_{\text{proj}}\}$
17: **end for**
18:
19: // Integrate genre information into visual features
20: **for** $i = 1$ **to** $N_t$ **do**
21:      genre_context $\leftarrow 0$
22:      **for** $j = 1$ **to** $N_g$ **do**
23:          genre_context $\leftarrow$ genre_context $+ A[i, j] \times G_{\text{projected}}[j]$
24:      **end for**
25:      $V_t^{\text{genre}}[i] \leftarrow V_t[i] + $ genre_context // additive integration
26: **end for**
27: **Output:** Genre-enhanced visual features $V_t^{\text{genre}} \in \mathbb{R}^{N_t \times D}$

---

Additionally, we provide the pseudo boundary generation flow incorporating our probabilistic anchor sampling approach that utilizes shot duration in pseudocode form.

---

**Algorithm 3** Probabilistic Anchor-based Pseudo Boundary Generation

---

1: **Input:** Shot encoder $f_{\text{enc}}$, shot sequence $S = \{s_1, s_2, \ldots, s_N\}$, sequence length $L = 17$
2: Initialize: anchor_left $\leftarrow \emptyset$, anchor_right $\leftarrow \emptyset$
3:
4: // Compute duration-based sampling weights
5: **for** $i = 1$ **to** $N$ **do**
6:    $w_i \leftarrow 1/d_i$ // inverse duration weighting
7: **end for**
8: $W \leftarrow [w_1/\sum w_i, w_2/\sum w_i, \ldots, w_N/\sum w_i]$ // normalize to probabilities
9:
10: // Probabilistic anchor sampling
11: anchor_left $\leftarrow$ sample from $S[: \text{N}//2]$ using weights $W[: \text{N}//2]$
12: anchor_right $\leftarrow$ sample from $S[\text{N}//2 + 1 :]$ using weights $W[\text{N}//2 + 1 :]$
13:
14: // Extract shot representations for subsequence
15: $E_{\text{subseq}} \leftarrow []$
16: **for** $i = $ anchor_left **to** anchor_right **do**
17:    $e_i \leftarrow \text{MLP}(f_{\text{enc}}(s_i))$ // project encoded features
18:    $E_{\text{subseq}} \leftarrow \{E_{\text{subseq}}; e_i\}$ // append
19: **end for**
20:
21: // Generate pseudo boundary using DTW alignment
22: $e_{\text{l}} \leftarrow E_{\text{subseq}}[0]$ // left anchor representation
23: $e_{\text{r}} \leftarrow E_{\text{subseq}}[-1]$ // right anchor representation
24: $b^* \leftarrow \text{DTW\_align}(e_{\text{l}}, e_{\text{r}}, E_{\text{subseq}})$ // find optimal boundary
25:
26: **Output:** Pseudo boundary position $b^*$ within subsequence [anchor_left, anchor_right]

---

| Genre | Prompt |
|---|---|
| Drama | Emotional expressions and intimate character interactions captured in natural lighting. |
| Comedy | Exaggerated character expressions and reactions highlighted with clear visibility. |
| Thriller | High contrast lighting and strategic framing creating tense atmospheric composition. |
| Action | Dynamic movement and high energy composition with spectacular visual impact. |
| Romance | Intimate character relationships emphasized through soft lighting and warm tones. |
| Horror | Unsettling visual elements with extreme shadows and distorted perspectives. |
| Crime | Urban environment details highlighted with stark contrast lighting. |
| Documentary | Authentic visual capture of real people in natural, unposed compositions. |
| Adventure | Expansive environmental views emphasizing scale and exploration. |
| Sci-Fi | Distinctive technological elements presented with artificial lighting and futuristic composition. |
| Family | Bright, warm colors and clear composition creating welcoming atmosphere. |
| Fantasy | Ethereal lighting and visually enhanced reality elements creating magical atmosphere. |
| Mystery | Selective focus and carefully hidden details in shadowed composition. |
| Biography | Documentary-style framing of actual person with period-specific details. |
| Animation | Stylized color palette and intentionally crafted visual elements in artistic composition. |
| History | Period-accurate visual elements presented in formal historical composition. |
| Music | Stage lighting and dynamic presentation of musical performance elements. |
| War | Military visual elements presented in desaturated colors with conflict intensity. |
| Sport | Physical action and competitive movement captured with clear focus. |
| Musical | Theatrical staging and choreographed elements with dramatic lighting. |
| Western | Frontier period elements portrayed with earthy tones in western composition. |

Table 17: Style prompts for each genre.

| Genre | Prompt |
|---|---|
| Drama | Should contain numerous consecutive scenes of characters portrayed to effect a serious narrative throughout the title, usually involving conflicts and emotions. This can be exaggerated upon to produce melodrama. Subjective. |
| Comedy | Virtually all scenes should contain characters participating in humorous or comedic experiences. The comedy can be exclusively for the viewer, at the expense of the characters in the title, or be shared with them. Please submit qualifying keywords to better describe the humor (i.e. spoof, parody, irony, slapstick, satire, dark-comedy, comedic-scene, etc.). Subjective. |
| Thriller | Should contain numerous sensational scenes or a narrative that is sensational or suspenseful. Note: not to be confused with Mystery or Horror, and should only sometimes be accompanied by one (or both). Subjective. |
| Action | Should contain numerous scenes where action is spectacular and usually destructive. Often includes non-stop motion, high energy physical stunts, chases, battles, and destructive crises (floods, explosions, natural disasters, fires, etc.) Note: if a movie contains just one action scene (even if prolonged, i.e. airplane-accident) it does not qualify. Subjective. |
| Romance | Should contain numerous inter-related scenes of a character and their personal life with emphasis on emotional attachment or involvement with other characters, especially those characterized by a high level of purity and devotion. Note: Reminder, as with all genres if this does not describe the movie wholly, but only certain scenes or a subplot, then it should be submitted as a keyword instead. Subjective. |
| Horror | Should contain numerous consecutive scenes of characters effecting a terrifying and/or repugnant narrative throughout the title. Note: not to be confused with Thriller which is not usually based in fear or abhorrence. Subjective. |
| Crime | Whether the protagonists or antagonists are criminals this should contain numerous consecutive and inter-related scenes of characters participating, aiding, abetting, and/or planning criminal behavior or experiences usually for an illicit goal. Not to be confused with Film-Noir, and only sometimes should be supplied with it. Subjective. |
| Documentary | Should contain numerous consecutive scenes of real personages and not characters portrayed by actors. This does not include fake or spoof documentaries, which should instead have the fake-documentary keyword. A documentary that includes actors re-creating events should include the keyword "reenactment" so that those actors are not treated as "Himself." This genre should also be applied to all instances of stand-up comedy and concert performances. Objective. |
| Adventure | Should contain numerous consecutive and inter-related scenes of characters participating in hazardous or exciting experiences for a specific goal. Often include searches or expeditions for lost continents and exotic locales, characters embarking in treasure hunt or heroic journeys, travels, and quests for the unknown. Not to be confused with Action, and should only sometimes be supplied with it. Subjective. |
| Sci-Fi | Numerous scenes, and/or the entire background for the setting of the narrative, should be based on speculative scientific discoveries or developments, environmental changes, space travel, or life on other planets. Subjective. |
| Family | Should be universally accepted viewing for a younger audience. e.g., aimed specifically for the education and/or entertainment of children or the entire family. Often features children or relates to them in the context of home and family. Note: Usually, but not always, complementary to Animation. Objective. |
| Fantasy | Should contain numerous consecutive scenes of characters portrayed to effect a magical and/or mystical narrative throughout the title. Usually has elements of magic, supernatural events, mythology, folklore, or exotic fantasy worlds.Note: not to be confused with Sci-Fi which is not usually based in magic or mysticism. Subjective. |
| Mystery | Should contain numerous inter-related scenes of one or more characters endeavoring to widen their knowledge of anything pertaining to themselves or others. Note: Usually, but not always associated with Crime. Subjective. |
| Biography | Primary focus is on the depiction of activities and personality of a real person or persons, for some or all of their lifetime. Events in their life may be reenacted, or described in a documentary style. If re-enacted, they should generally follow reasonably close to the factual record, within the limitations of dramatic necessity. A real person in a fictional setting would not qualify a production for this genre. Objective. |
| Animation | Over 75% of the title's running time should have scenes that are wholly, or part-animated. Any form of animation is acceptable, e.g., hand-drawn, computer-generated, stop-motion, etc. Puppetry does not count as animation, unless a form of animation such as stop-motion is also applied. Incidental animated sequences should be indicated with the keywords part-animated or animated-sequence instead. Objective. |
| History | Primary focus is on real-life events of historical significance featuring real-life characters (allowing for some artistic license); in current terms, the sort of thing that might be expected to dominate the front page of a national newspaper for at least a week; for older times, the sort of thing likely to be included in any major history book. Objective. |
| Music | Contains significant music-related elements while not actually being a Musical; this may mean a concert, or a story about a band (either fictional or documentary). Subjective. |
| War | War, a film genre, Should contain numerous scenes and/or a narrative that pertains to a real war (i.e., past or current). Note: for titles that portray fictional war, please submit it as a keyword only. Objective. |
| Sport | Sport, a film genre, Focus is on sports or a sporting event, either fictional or actual. This includes fictional stories focused on a particular sport or event, documentaries about sports, and television broadcasts of actual sporting events. In a fictional film, the sport itself can also be fictional, but it should be the primary focus of the film. |
| Musical | Should contain several scenes of characters bursting into song aimed at the viewer (this excludes songs performed for the enjoyment of other characters that may be viewing) while the rest of the time, usually but not exclusively, portraying a narrative that alludes to another Genre. Note: not to be added for titles that are simply music related or have music performances in them; e.g., pop concerts do not apply. Also, classical opera, since it is entirely musical, does not apply and should instead be treated as Music. Objective. |
| Western | Should contain numerous scenes and/or a narrative where the portrayal is similar to that of frontier life in the American West during 1600s to contemporary times. Objective. |

Table 18: Definition prompts for each genre.

