# OpenReview forum: "Video Scene Segmentation with Genre and Duration Signals"
_ICLR.cc/2026/Conference — ICLR 2026 Poster_

### Official Review · Reviewer_NJjj · 2025-10-29

**Soundness:** 3
**Presentation:** 3
**Contribution:** 2
**Rating:** 6
**Confidence:** 3

**Summary:**

This paper proposes a metadata-guided framework for video scene segmentation. Instead of relying only on visual similarity between adjacent shots, the authors incorporate genre conventions derived from IMDb textual definitions as semantic priors into a ViT-based shot encoder through affinity-based residual fusion, which functions similarly to cross-attention. They further introduce inverse-duration-weighted anchor sampling to improve pseudo-boundary generation during self-supervised pretraining and a test-time shot splitting strategy to handle long shots without retraining. In addition, they present a new benchmark, MovieChat-SSeg, with 1,000 manually annotated clips covering movies, TV series, and documentaries. Experiments on MovieNet-SSeg, BBC, and MovieChat-SSeg datasets show consistent improvements over previous methods such as BaSSL and TranS4mer. The authors also provide comprehensive ablations covering integration strategies, prompt designs, anchor sampling, and split thresholds.

**Strengths:**

1.The idea of incorporating genre metadata as soft priors is well motivated and practical.

2.The method is lightweight and can be easily applied to existing models.

3.Experimental results show consistent improvement across multiple benchmarks and good zero-shot generalization.

4.Ablation studies are detailed and analyze four aspects: genre integration, prompt strategy, anchor sampling, and shot split thresholds.

5.The proposed MovieChat-SSeg benchmark adds valuable evaluation data with diverse narrative structures.

**Weaknesses:**

1. The novelties of the proposed method could be highlighted by comparing with the most related methods, e.g. whether these genre and duration signals are newly proposed ? What are exactly advantages of these characteristics over the existing ones, e.g. video context.
[1] Video Scenes Segmentation Based on Multimodal Genre Prediction, Procedia Computer Science, 2020.

2. Robustness to noisy or conflicting genre priors is not explored. There is no experiment showing how the model behaves when the genre is wrong or uncertain.

3.Multi-genre scenarios are not addressed in detail, which may lead to bias or performance drop in real applications.

4.The efficiency impact of test-time shot splitting is not reported. This makes it difficult to assess its practicality for deployment.

5.Failure case analysis is weak and the qualitative examples focus mostly on positive results.

6. Although the state of the arts are compared in Table 9 of appendix, the comparison in Table 2 should include the methods published in recent two years to validate the superiority of the proposed algorithm.

7. Typo errors, e.g. `a a scene` in line 162.

**Questions:**

1.The novelties and advantages of the proposed genre and duration signals could be highlighted.

2.How does the model behave if the genre prior is noisy or incorrect? Could you provide degradation curves or sensitivity analysis?

3.Is there statistical evidence that shorter shots are more likely to be boundaries? Could you provide quantitative analysis?

4.How do you handle multi-genre videos during training and inference?

5. What is the computational overhead of test-time shot splitting?

6.Could you show more failure case analysis, especially on non-narrative content such as news or user-generated videos?

7. Are there any related state of the arts that are published in recent two years for the comparison in Table 2 ?

---

> ### Author Response · Authors · 2025-11-20
>
> Thank you for your constructive feedback. We address your concerns as follows:
> Changes related to your review are marked in blue in the revised manuscript.
>
> ---
>
> **W1 & Q1: Novelty of genre and duration signals**
>
> We have added comparison with prior genre-based methods in our revised Related Work section.
> VSS-MGP (2020) requires shot-level genre predictions, while Movie-CLIP (2024) does not explicitly use genre for segmentation.
> We clarify our distinct approach eliminates shot-level genre classification by integrating genre embeddings as semantic cues and leveraging duration for self-supervised learning.
>
> While BaSSL and TranS4mer advance pseudo-boundary learning through improved
> visual representations, our work introduces orthogonal innovations across
> representation, sampling, and inference:
>
> | **Component** | **BaSSL** | **TranS4mer** | **Ours** | **Key Differences & Innovations** |
> |---------------|-----------|---------------|----------|-----------------------------------|
> | **Shot Representation Learning** | Visual features only (ResNet-50) | Visual features (ViT with state-space model) | **Genre-guided visual features** | **First to use genre as semantic cue**: Integrates textual genre definitions into shot-level representation learning via CLIP embeddings |
> | **Pseudo-Boundary Generation** | Fixed anchors (first/last shots) | Fixed anchors strategy | **Duration-aware anchor sampling** | **Leverages shot duration patterns**: Probabilistic sampling based on inverse-duration weighting for better pseudo-boundaries |
> | **Test-Time Processing** | Standard inference | Standard inference | **Adaptive shot splitting** | **Handles long-duration shots**: Splits shots exceeding 10s threshold at test time, adding 16.6% overhead for 0.18 AP gain |
> | **Dataset Contribution** | - | - | **MovieChat-SSeg (cross-domain)** | **New diverse benchmark**: 1K videos spanning movies/TV/documentaries, to be released publicly |
>
>
> **W2, W3, Q2 & Q4: Robustness and multi-genre handling**
>
> We provide robustness analysis (Appendix C, Tables 9).
> Our architecture uses a fixed genre embedding pool as non-trainable parameters in the shot encoder, shared across all videos rather than video-specific inputs.
> Genre vocabulary reduction experiments (21, 16, 11, 6 genres) show graceful degradation with interesting patterns—6 genres outperform 11, suggesting removing additional niche genres eliminates conflicting priors. Even with minimal vocabulary (6 genres, ~30\%), performance maintains +2.14 AP over genre-free baseline.
> Multi-genre videos are naturally accommodated through soft integration via affinity mapping.
> Analysis across 1-5 genres per video (Appendix C, Table 10) shows monotonic degradation reflecting increased narrative complexity.
>
> **W4, Q3 & Q5: Shot duration strategy and computational cost**
>
> We clarify our training strategy: shorter shots serve as anchors for pseudo-boundary identification in self-supervised learning (Appendix D, Figure 8), not as assumed boundaries.
> This enables diverse transition points for improved representation learning.
> We provide computational cost analysis (Appendix E, Table 12): test-time splitting at 10-second threshold splits 8.3% of shots, adding ~16.6% overhead for +0.18 AP.
> Higher thresholds (30s: 2.0%; 60s: 0.4%) reduce cost with diminished gains.
>
> **W5, W6, Q6 & Q7: Failure analysis and recent comparisons**
>
> We have added comprehensive failure case analysis (Section 5.6, Figure 3) on the three lowest-performing videos, identifying five systematic error patterns including insert shot misclassification, lighting-induced false positives, missed spatial transitions, extreme framing artifacts, and special effects handling.
> This analysis reveals inherent limitations of visual-only approaches in distinguishing stylistic variations from semantic boundaries.
> We have also updated Table 2 to include recent methods (2023-2024), demonstrating competitive performance against latest approaches.
>
> ## Summary
>
> Key additions: genre vocabulary robustness (Appendix C, Tables 9), multi-genre analysis (Appendix C, Table 10), shot duration visualization (Appendix D, Figure 8), computational cost (Appendix E, Table 12), failure case analysis (Section 5.6, Figure 3), updated SOTA comparisons (Table 2). We have also corrected the typo and performed thorough proofreading throughout the manuscript.

---

> > ### Comment · Reviewer_NJjj · 2025-11-26
> > **I will maintain my original score.**
> >
> > Thanks for the rebuttal. Most of my concerns have been well addressed, while the compared methods published in 2025 could be further appended. Thus, I will maintain my original score.

---

> > > ### Author Response · Authors · 2025-12-01
> > >
> > > Thank you for the constructive feedback on including recent 2025 methods.
> > > We have now added the SceneTiling [1] result to Table 2.
> > > Below is a condensed comparison showing the top-performing methods from each learning paradigm (unsupervised, supervised, and self-supervised):
> > >
> > > | Method | AP | AUC | F1 | mIoU |
> > > |--------|------|------|------|------|
> > > | **Unsupervised** | | | | |
> > > | SceneTiling (Wang et al. (2025)) | 19.95 | 72.15 | 29.36 | 36.96 |
> > > | Story Graph (Tapaswi et al. (2014)) | 25.10 | - | - | 35.70 |
> > > | Grouping (Rotman et al. (2016)) | 33.60 | - | - | 37.20 |
> > > | **Supervised** | | | | |
> > > | LGSS (Rao et al. (2020)) | 47.10 | - | - | 48.80 |
> > > | Movie-CLIP (Zhang et al. (2024)) | 54.45 | - | - | - |
> > > | MHRT (Wei et al. (2023)) | 54.80 | 90.30 | 46.30 | 51.20 |
> > > | **Self-supervised** | | | | |
> > > | CAT (Yang et al. (2023)) | 59.55 | 91.81 | 51.93 | 53.67 |
> > > | TranS4mer (Islam et al. (2023)) | 60.78 | 91.89 | 48.36 | 51.91 |
> > > | **Ours** | **63.62** | **92.84** | **58.88** | **59.64** |
> > >
> > > **(a) Results on the MovieNet-SSeg.**
> > >
> > >
> > > **SceneTiling Implementation & Results**
> > > We reimplemented SceneTiling from VideoLLaMB (ICCV 2025)[1], originally proposed for partitioning video sequences into semantic units for video QA tasks. Their approach detects semantic transitions by analyzing representation similarity distributions between adjacent sampled frames to create meaningful sub-sequences. When applied to scene segmentation benchmarks, the zero-shot evaluation shows competitive performance with existing unsupervised methods, indicating that semantic transition detection approaches from video understanding can be effectively adapted for scene boundary detection tasks.
> > > We believe this addition strengthens our comparison with recent long-context video understanding methods. We appreciate your guidance in making our work more comprehensive.
> > >
> > > [1] VideoLLaMB: Long Streaming Video Understanding with Recurrent Memory Bridges, ICCV, 2025

---

### Official Review · Reviewer_KJQR · 2025-11-01

**Soundness:** 3
**Presentation:** 3
**Contribution:** 3
**Rating:** 6
**Confidence:** 4

**Summary:**

The authors target the problem of scene segmentation for long form videos. They propose exploiting the genre information to enhance the semantic information to learn better shot representations during the pretraining stage. They do this by constructing genre-level textual prompts and encode them into embedding vectors and then use the relevance of each genre to the given visual token an affinity matrix and add it to the visual features. Furthermore the authors also propose anchor sampling scheme that gives more importance to shorter duration shots as against just using the sequence endpoints. At the test time, the authors further counter the high variability of the shot durations by just splitting the sho greater than certain threshold duration into three shots and use uniform sampling to extract the keyframes. Also, the authors propose a new manually annotated dataset which spans movies, TV series and documentaries and thus consists of sufficiently wide variety of inputs. For the pretraining stage the authors just use a standard contrastive loss annd also another cross entrop loss to distinguish a boundary shot from the non-boudary counterpart and use also the CE loss for finetuning. The authors finally reporrt the resuls on MovieNet SSeg for supervised training and evaluate the generalization on their datasets showing significant gains in Average precision for both and F1 metric for the first.

**Strengths:**

The overall proposed scheme for training including Genre guided shot representation and the inverse duration shot sampling seems to be sound semantically since in long videos the genre based prompts can help in the boundary detection. The test-time shot splitting strategy is standard and a sound way to handle the long videos considering variability in the shot duration. The results also sucessfully second that. The proposed dataset can also be used to drive the research further in this direction.

**Weaknesses:**

The proposed scheme doesn't seem to be very novel and can be more interpreted as standard way-out for solving this long-form video issue in the considered setup. The shot sampling although is a good finding but I am unsure of any novelty including the test time splitting. So method wise not a major contribution although results seem interesting.

Initially I have given a rating 6 due to the impact solving the problem can make, but I might change it based on other reviews and the responses authors provide.

**Questions:**

When comparing with other baselines do the authors incorporate the textual information since they mentioned that they reimplent the BaSSL method. Or they just do it to avoid architectural differences? If they don't use the textual information while comparing then isn't it unfair to them?

---

> ### Author Response · Authors · 2025-11-20
>
> Thank you for your thoughtful review. We appreciate your recognition of our results and understand your concerns about novelty.
> We address the technical contributions of our approach and the comparison question below.
>
> ---
>
> ## Technical Contributions
>
> While existing methods like BaSSL and TranS4mer have established strong foundations in video scene segmentation, we would like to clarify the specific innovations introduced to address four fundamental challenges:
> (1) **semantic representation learning beyond visual cues**,
> (2) **effective pseudo-boundary generation using cinematic principles**,
> (3) **adaptive shot splitting at inference**,
> and (4) **new benchmark dataset across diverse video content types**.
> Our contributions go significantly beyond conventional approaches by integrating production-level metadata into the learning framework.
>
> The table below clearly highlights the key differences:
>
> | **Component** | **BaSSL** | **TranS4mer** | **Ours** | **Key Differences & Innovations** |
> |---------------|-----------|---------------|----------|-----------------------------------|
> | **Shot Representation Learning** | Visual features only (ResNet-50) | Visual features (ViT with state-space model) | **Genre-guided visual features** | **First to use genre as semantic cue**: Integrates textual genre definitions into shot-level representation learning via CLIP embeddings |
> | **Pseudo-Boundary Generation** | Fixed anchors (first/last shots) | Fixed anchors strategy | **Duration-aware anchor sampling** | **Leverages shot duration patterns**: Probabilistic sampling based on inverse-duration weighting for better pseudo-boundaries |
> | **Test-Time Processing** | Standard inference | Standard inference | **Adaptive shot splitting** | **Handles long-duration shots**: Splits shots exceeding 10s threshold at test time, adding 16.6% overhead for 0.18 AP gain |
> | **Dataset Contribution** | - | - | **MovieChat-SSeg (cross-domain)** | **New diverse benchmark**: 1K videos spanning movies/TV/documentaries, to be released publicly |
>
>
> Our approach advances beyond standard metadata usage through specific architectural and methodological designs.
>
> **Genre embedding architecture.**
> We derive genre embeddings from textual definitions and integrate them as fixed parameters within the shot encoder,
> where they interact with visual tokens through affinity mapping.
> This differs fundamentally from VSS-MGP (2020), which requires shot-level genre classification,
> and Movie-CLIP (2024), which does not explicitly leverage genre information.
> Our design eliminates per-shot annotation requirements while enabling semantic guidance through a shared 21-genre embedding pool across all videos.
>
> **Duration-based anchor sampling.**
> Instead of standard fixed-endpoint anchors, we use duration-weighted sampling of shorter shots for pseudo-boundary identification during self-supervised pre-training (Appendix D, Figure 8).
> This strategy creates diverse training segments, improving representation quality as demonstrated through ablation.
> We document computational implications (Appendix E, Table 12), showing our 10-second threshold adds 16.6% overhead for 0.18 AP gain.
>
> **Empirical rigor.**
> We examine robustness across reduced genre vocabularies (21→6 genres, Table 9), analyze multi-genre video performance (Table 10),
> and identify systematic failure patterns through lowest-performing video analysis (Section 5.6, Figure 3).
> These investigations reveal both capabilities and limitations of our visual-centric design.
>
> ---
>
> ## Q: Textual Information in Baseline Comparisons
>
> Our BaSSL reimplementation serves a specific purpose: demonstrating that
> our performance gains result from our proposed methods, not simply from
> using a different backbone or pretrained weights. We precisely follow
> BaSSL's original visual-only architecture but replace ResNet-50 with our
> ViT backbone and CLIP pretrained weights to ensure fair comparison under
> identical visual feature settings.
>
> Table 3 shows our reimplemented BaSSL achieves substantial improvement
> over the original. However, our full method still outperforms this strong
> baseline by 3.22 AP. This gap directly highlights our contribution:
> integrating genre embeddings as semantic cues and leveraging duration for
> anchor sampling. By controlling the backbone and pretrained weights, we
> isolate the effect of our metadata-driven innovations.
>
> BaSSL's original design does not incorporate genre and duration metadata
> available in MovieNet. Our ablation studies (Table 3) quantify each
> component's contribution, separating metadata effects from architectural
> choices. This comparison framework is standard practice: we evaluate our
> proposed integration methods against baselines using their original designs
> while controlling for implementation variations.

---

> ### Comment · Reviewer_KJQR · 2025-11-25
>
> Thanks for a detailed response! I am still not confident on the technical novelty part. Regarding the baseline question, I acknowledge the reviewers response and will maintain my original rating.

---

> > ### Author Response · Authors · 2025-11-26
> >
> > Thanks for your comments which make our paper polish up. If you have any question or idea to address your concern on the technical novelty, please let us know.

---

### Official Review · Reviewer_DYd7 · 2025-11-02

**Soundness:** 3
**Presentation:** 3
**Contribution:** 3
**Rating:** 6
**Confidence:** 4

**Summary:**

This paper introduces a novel method for video scene segmentation, which extends beyond the traditional reliance on visual similarity between consecutive shots by leveraging production-level metadata in the form of genre conventions and shot duration patterns. Scene segmentation is designed to identify semantically consistent boundaries within long-form videos, thus linking low-level visual cues with broader narrative comprehension. The principal contribution of this work is the inclusion of narrative-oriented signals into a self-supervised learning framework, which  improves shot-level representation learning.

**Strengths:**

*  Integration of contextual movie level metadata: Inclusion of textual genre level definitions as semantic priors for guiding shot level representations.
* Duration based pseudo boundary generation: Inverse duration-based anchor sampling strategy for providing more weights to shorter shots as compared to fixed anchor approaches like BaSSL
* Introduction of a benchmark for visual scene segmentation called MovieChat-SSeg associated with domains like movies, TV series and documentaries.
* State of the art performance on MovieNet-SSeg dataset when compared to other state-of-the-art self-supervised methods like TranS4mer and BaSSL.

**Weaknesses:**

* Non-visual cues like background music and audio events help identify scene divisions but are not included.
* Genre based contextual information might be insufficient in dialogue heavy scenes with minimal visual variations.
* The final ablation studies do not indicate whether performance varies by genre class. For instance, including the definition of an action genre may affect results more than drama.
* Although the proposed method has been evaluated on movies, TV series, and documentaries, it can also be applied to other formats such as news broadcasts, vlogs, and educational videos.

**Questions:**

* What additional forms of production metadata—apart from genre, duration, shot scale, and shot angle—could be utilised as semantic priors, and how might their integration differ in terms of complexity and potential impact compared to primary signals such as genre and duration? Possible examples include camera movement classifications (e.g., panning vs. static), or color palettes associated with mood shifts or emotional cues associated with character.

* Do shorter video segments (avg. 7.4 min in MovieChat-SSeg) limit the model's ability to capture long-term narrative patterns in full-length content?

---

> ### Author Response · Authors · 2025-11-20
>
> Thank you for your detailed and constructive review.
> We have carefully considered all your comments and made substantial revisions accordingly.
> Our point-by-point responses are provided below.
> Changes related to your review are marked in red in the revised manuscript.
>
> ---
>
> ## Weaknesses
>
> **W1 & W2: Multi-modal features (audio/dialogue) for scene segmentation**
>
> We quantify visual-only limitations through dialogue density analysis (Table 14, Appendix G): AP drops from 76.92 (low dialogue) to 61.62 (high dialogue). While genre metadata provides valuable narrative priors, this performance gap confirms that dialogue-heavy scenes require audio/speech integration—an important direction for future work.
>
> **W3: Missing per-genre performance analysis**
>
> Genre-specific analysis (Table 15, Appendix G) shows test-time split benefits Mystery/Crime genres where longer shots enable finer boundary detection, while Adventure/War genres show limited or negative impact due to already-short shots from rapid pacing. This validates our method's sensitivity to genre-specific shot characteristics.
>
> **W4: Applicability to non-narrative formats**
>
> Investigation on LongerVideos dataset (Figure 10, Appendix H) reveals fundamental limitations on lectures and entertainment content. Failures highlight that non-narrative "scenes" differ fundamentally, requiring redefined boundaries and audio integration—valuable future work beyond our narrative-focused scope.
>
> ---
>
> ## Questions
>
> **Q1: Additional metadata types**
>
> Experiments with camera movement (63.24 AP) and color grading (63.57 AP) are added in Appendix C, Table 7. Camera movement is limited by our keyframe-based representation, while color grading achieves comparable performance. These experiments reveal that effective metadata integration requires careful assessment of semantic relevance to scene boundaries, as camera movement relates more to shot-level cinematography while color grading shows scene-level potential.
>
> **Q2: Segment length limitations**
>
> 7.4-minute segments in MovieChat-SSeg provide sufficient context, as scene boundaries are local phenomena. Our segments contain an average of 116.4 shots across 7.8 scenes, exceeding SOTA's 17-25 shot windows by 4-6 times, and competitive benchmark performance validates this approach. While full-length context might offer benefits similar to human narrative understanding, computational costs and uncertain gains for shorter videos suggest this as future work rather than a fundamental limitation.
>
> ---
>
> ## Summary
>
> Key additions: dialogue density analysis (Table 14, Appendix G), genre-specific analysis (Table 15, Appendix G), additional metadata experiments (Table 7, Appendix C), non-narrative results (Figure 10, Appendix H). We hope these improvements address your concerns.

---

### Meta-Review · Area_Chair_AEfW · 2026-01-05

**Summary:**

This submission proposes a metadata-guided self-supervised framework for long-form video scene segmentation, incorporating genre conventions (textual prompts) and shot duration patterns as priors to improve shot representations and pseudo-boundary generation. All reviewers agree the paper is sound, well-presented, and demonstrates strong empirical performance, including a new benchmark dataset (MovieChat-SSeg).

The main concerns are on the technical novelty, scope limitations, and analysis gaps. The rebuttal was strong and addressed many points by adding genre breakdowns, robustness experiments, cost analysis, and failure cases. Initial ratings from the three reviewers are 6, 6, and 6. So there is consensus. AC has checked the submission and the reviews and agreed with the consensus, thus acceptance is recommended.

**Reviewer Concerns:**

Most of the concerns have been addressed.

**Reviewer Scores:**

Initial 6, 6, 6. After the rebuttal, most likely they remain 6, 6, 6.

---

### Decision · Program_Chairs · 2026-01-26

Accept (Poster)